# BN-Pool: Bayesian Nonparametric Pooling for Graphs

**Daniele Castellana**                                                     *daniele.castellana@unifi.it*
*Università degli Studi di Firenze*

**Filippo Maria Bianchi**                                                  *filippo.m.bianchi@uit.no*
*UiT The Arctic University of Norway* and
*NORCE Norwegian Research Centre AS*

**Reviewed on OpenReview:** *https: // openreview. net/ forum? id= 3B3Zr2xfkf*

## Abstract

We introduce BN-Pool, the first clustering-based pooling method for Graph Neural Networks that adaptively determines the number of supernodes in a coarsened graph. BN-Pool leverages a generative model based on a Bayesian nonparametric framework for partitioning graph nodes into an unbounded number of clusters. During training, the node-to-cluster assignments are learned by combining the supervised loss of the downstream task with an unsupervised auxiliary term, which encourages the reconstruction of the original graph topology while penalizing unnecessary proliferation of clusters. By automatically discovering the optimal coarsening level for each graph, BN-Pool preserves the performance of soft-clustering pooling methods while avoiding their typical redundancy by learning compact pooled graphs. The code is available at https://github.com/NGMLGroup/Bayesian-Nonparametric-Graph-Pooling.

## 1 Introduction

Graphs sit at the heart of drug-discovery pipelines, traffic-flow simulators, social-network recommenders, and a growing list of web-scale systems. Thanks to Graph Neural Networks (GNNs), a powerful class of deep learning models designed to process graph-structured data, the state-of-the-art on those tasks has significantly improved over the past few years (Zhou et al., 2020). Despite the numerous advances in their architectural design, GNNs still struggles to learn hierarchical representations that are compact and consistently optimal for a wide range of downstream tasks.

Pooling, a fundamental component in computer vision architectures, becomes far trickier on irregular, non-Euclidean graphs, which are hard to down-sample without sacrificing structure or features. Popular and best-performing graph pooling operators (Wang et al., 2024) build coarse graphs by clustering nodes, but they hard-code the number of supernodes $K$ for every input graph (Ying et al., 2018; Bianchi et al., 2020a); thus, all the coarsened graphs have the same size. Moreover, tuning the value of $K$ can be difficult: rather than employing an expensive hyperparameter sweep, the common approach is to set it to a value large enough to avoid information loss (e.g., a fraction of the average size of all the graphs in the dataset). This rigidity prevents the model from adapting dynamically to the graph structure and produces redundant and dense representations (see Figure 1), which are less interpretable and yield unnecessary computations.

To overcome these limitations, we introduce Bayesian Nonparametric Pooling (BN-Pool), a novel graph pooling operator based on a Bayesian Nonparametric (BNP) technique. We define a generative process for the adjacency matrix of the input graph where the probability of having a link between two nodes depends on their cluster membership, thus ensuring that clusters reflect the graph topology. The BNP approach allows the number of clusters $K$ to adapt to each input graph, rather than being fixed in advance. Within our Bayesian framework, the clustering function is the posterior of the cluster membership given the input graph. We approximate the posterior by employing a GNN; on the one hand, this permits capturing complex relations that appear between the hidden and the observable variables; on the other hand, we can jointly condition the posterior on the graph topology, the node features, and the downstream task. The GNN parameters are

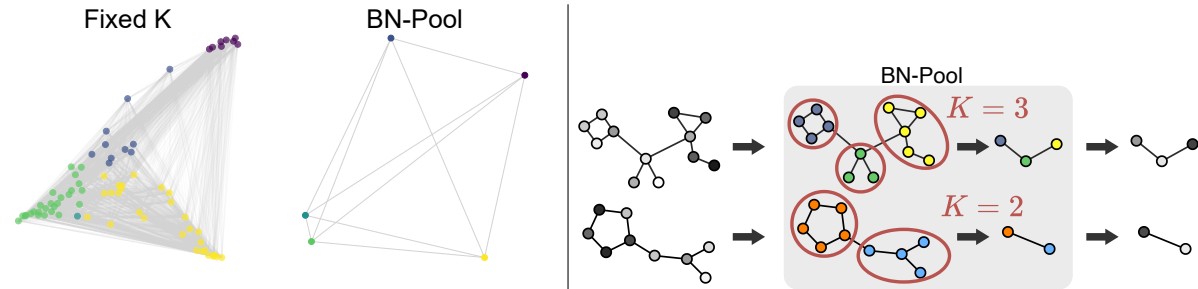

Figure 1: (**Left**) A typical pooled graph computed by a clustering-based pooling approach with fixed $K$ (left) and by BN-Pool (right). (**Right**) BN-Pool learns end-to-end the number of clusters $K$ to pool each graph independently.

trained by optimizing two complementary objectives: one defined by the loss of the downstream task (e.g., cross-entropy in graph classification), the other defined by an unsupervised auxiliary loss that derives from the probabilistic generative process.

Our main contributions are:

- We introduce the first soft-clustering pooling operator capable of adaptively determining the number of supernodes for each input graph.

- We adapt the Stochastic Variational Inference (SVI) framework to a training procedure that enables seamless integration of BNP with GNN architectures and supports end-to-end optimization.

- We validate the effectiveness of our approach on both node clustering and graph classification tasks. In the former, our method successfully identifies communities and their interaction patterns. In the latter, it achieves performance on par with or superior to existing pooling methods, demonstrating the ability to generate compact graph representations without compromising informative content.

The paper is organized as follows: in Section 2, we introduce the preliminary concepts relevant to our work, namely the Dirichlet Process (DP), GNNs, and graph pooling operators. In Section 3, we present the proposed methodology by defining the generative process, the training procedure, and the interpretation of the model's hyperparameters. Section 4 discusses existing approaches that are related to our work, while Section 5 presents the results obtained on both node clustering and graph classification tasks. Finally, in Section 6, we conclude and outline possible directions for future work.

## 2 Preliminaries

### 2.1 Bayesian Nonparametric and Dirichlet Process

The BNP framework (Orbanz & Teh, 2010) aims to build nonparametric models by applying Bayesian techniques. The term *nonparametric* indicates the ability of a model to adapt its size (i.e., the number of parameters) directly to data. In contrast, in the *parametric* approach, the model size is fixed in advance by setting some hyperparameters.

The BNP literature most relevant to our work relates to the family of Dirichlet Process (DP) (Gershman & Blei, 2012). In its most essential definition, a DP is a stochastic process whose samples are categorical distributions of infinite size. Thus, in the same way as the Dirichlet distribution is the conjugate prior for the categorical distribution, the DP is the conjugate prior for infinite discrete distributions. A classical use of the DP is in the definition of mixture models that allow an infinite number of components, where the DP is used as the prior distribution over the mixture weights. A key property of the DP is its clustering behavior: even if there is an infinite number of components available, the DP tends to reuse the components that have already been used.

Let $G_0$ be a continuous distribution, and let $\alpha_{\mathrm{DP}}$ be a positive real number. We write:

$$G \sim \mathrm{DP}(\alpha_{\mathrm{DP}}, G_0), \tag{1}$$

where $G$ is a discrete distribution with the same support as $G_0$, meaning that the probability of two samples of $G$ being equal is non-zero, but has a countably infinite number of point masses. Figure 2 shows an example of three different draws of $G$ when the base distribution $G_0$ is a skewed Normal, and the value of $\alpha_{\mathrm{DP}}$ is 10, 100, and 1000. As we can see from the figure, $G$ represents a discrete approximation of $G_0$, where the concentration parameter $\alpha_{\mathrm{DP}}$ indicates how much the mass in $G$ is concentrated around a given point; the base distribution is the expected value of the process, i.e., the DP draws distributions around the base distribution $G_0$, much like a normal distribution draws real numbers around its mean.

The DP clustering property does not emerge from the previous formulation, which also does not tell us how to compute $G$. In the following, we describe the *Blackwell-MacQueen urn scheme* (Blackwell & MacQueen, 1973) and the *stick breaking process* (Sethuraman, 1994). While the former provides a good intuition of the clustering property of a DP, the latter offers a constructive formulation that we leverage in this work.

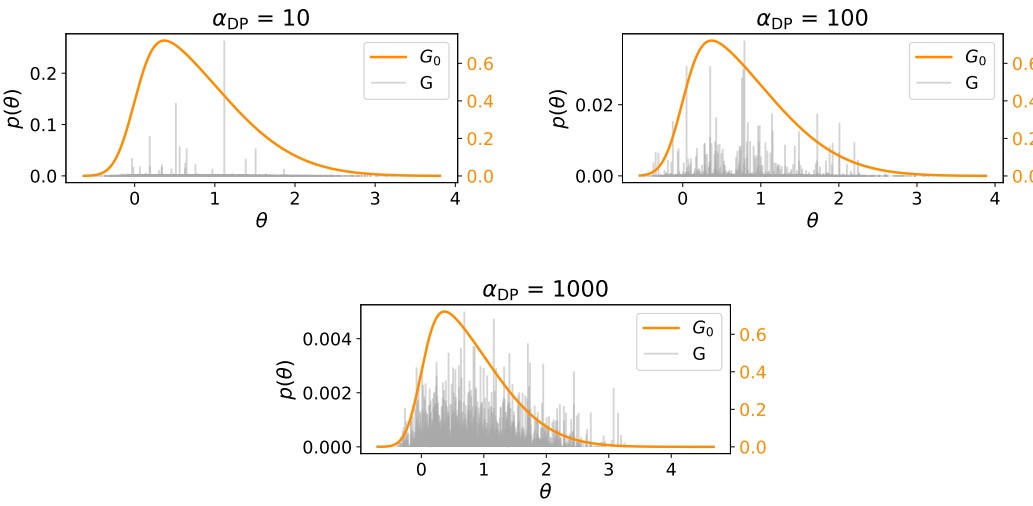

Figure 2: Three single draws from the DP using as $G_0$ a Normal skewed distribution and three different $\alpha_{\mathrm{DP}}$ values. Note that each plot has a different scale on the $y$-axis.

**Blackwell-MacQueen urn scheme.** Let us consider a sequence of i.i.d. random variables $\theta_1, \theta_2, \ldots$ that are distributed according to $G \sim DP(\alpha_{\mathrm{DP}}, G_0)$. We can interpret the conditional distributions of $\theta_i$ given the previous $\theta_1, \ldots, \theta_{i-1}$, where $G$ has been integrated out, as a simple urn model containing balls with distinct colors (Blackwell & MacQueen, 1973). The balls are drawn equiprobably; when a ball is drawn, it is placed back in the urn together with another ball. The color of the new ball is identical to the color of the drawn ball with probability proportional to the number of balls of that color already in the urn; otherwise, with probability proportional to $\alpha_{\mathrm{DP}}$, we choose a new color drawn from $G_0$. This model exhibits a positive reinforcement effect: the more a color is drawn, the more likely it is to be drawn again.

Let $\phi_1, \ldots, \phi_K$ be the distinct atoms drawn from $G_0$ (i.e., the colors) that can be taken by $\theta_1, \ldots, \theta_{i-1}$ (i.e., the balls), and let $m_k$ be the number of times the atom $\phi_k$ appears in $\{\theta_1, \ldots, \theta_{i-1}\}$. Formally, we can express the sampling procedure as:

$$\theta_i \mid \theta_1, \ldots, \theta_{i-1} = \begin{cases} \phi_k \text{ with probability } \frac{m_k}{i-1+\alpha_{\mathrm{DP}}} \\ \text{a new draw from } G_0 \text{ with probability } \frac{\alpha_{\mathrm{DP}}}{i-1+\alpha_{\mathrm{DP}}} \end{cases} \tag{2}$$

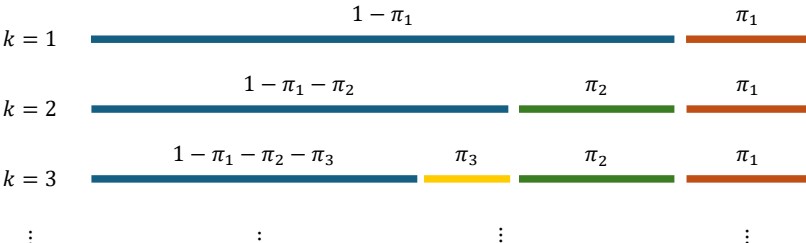

Figure 3: Graphical representation of the stick-breaking process.

Equivalently, we can write:

$$\theta_i \mid \theta_1, \ldots, \theta_{i-1} \sim \sum_{k=1}^{K} \frac{m_k}{i-1+\alpha_{\text{DP}}} \delta_{\phi_k} + \frac{\alpha_{\text{DP}}}{i-1+\alpha_{\text{DP}}} G_0, \tag{3}$$

where $\delta_{\phi_k}$ is the Dirac measure concentrated at $\phi_k$.

Referring to Figure 2, the values $m_k$ are proportional to the heights of the grey bars. When $\alpha_{\text{DP}}$ is small, most of the probability mass is concentrated in a few points. While the Blackwell-MacQueen urn scheme helps to understand the clustering property of DP, the sampling procedure does not provide an analytic expression of $G$ that can be exploited.

**Stick-breaking process.** The idea of the Stick-Breaking Process (SBP) (Sethuraman, 1994) is to repeatedly break off a "stick" of initial length 1. Each time we need to break the stick, we choose a value between 0 and 1 that determines the fraction we take from the remainder of the stick. In Figure 3, we show the iterative breaking process, where the values of $\pi_1, \pi_2, \pi_3, \ldots$ represent the parts of the stick pieces broken in the first three iterations.

Formally, the stick-breaking construction is based on independent sequences of i.i.d. random variables $(\pi'_k)_{k=1}^{\infty}$:

$$\pi'_k \mid \alpha_{\text{DP}} \sim \text{Beta}(1, \alpha_{\text{DP}}) \qquad\qquad \pi_k = \pi'_k \prod_{l=1}^{k-1} (1 - \pi'_l), \tag{4}$$

where the value of $\pi'_k$ indicates the proportion of the remaining stick that we break at iteration $k$. To understand the stick analogy, we should first convince ourselves that the quantity $\prod_{l=1}^{k-1}(1-\pi'_l)$ is equal to the length of the remainder of the stick $1 - \sum_{l=1}^{k-1} \pi_l$ after breaking it $k-1$ times. Thus, the length of the stick's piece $\pi_k$ is obtained by multiplying the stick fraction $\pi'_k$ by the length of the remaining stick $\prod_{l=1}^{k-1}(1-\pi'_l)$ at the $k$-th step.

It is important to note that the sequence $\boldsymbol{\pi} = (\pi_k)_{k=1}^{\infty}$ constructed by Equation 4 satisfies $\sum_{k=1}^{\infty} \pi_k = 1$ with probability one (Sethuraman, 1994). Thus, we may interpret $\boldsymbol{\pi}$ as a random probability measure on the positive integers. This distribution is often denoted as GEM, which stands for Griffiths, Engen, and McCloskey – see (Pitman, 2002).

Now we have all the ingredients to define a random measure $G \sim \text{DP}(\alpha_{\text{DP}}, G_0)$:

$$\phi_k \mid G_0 \sim G_0 \qquad\qquad G = \sum_{k=1}^{\infty} \pi_k \delta_{\phi_k}, \tag{5}$$

where $(\phi_k)_{k=1}^{\infty}$ are the atoms drawn from $G_0$ and $\delta_{\phi_k}$ is the Dirac measure concentrated at $\phi_k$. Sethuraman showed that $G$ as defined in Equation 5 is a random probability measure distributed according to $\text{DP}(\alpha_{\text{DP}}, G_0)$ (Sethuraman, 1994). The stick-breaking process is related to the urn scheme since the length of each piece $\pi_k$ corresponds to the expected probability of drawing a ball of color $\phi_k$ from the urn.

(a) Flat GNN.

(b) Hierarchical GNN.

Figure 4: **(a)** A sketch of a "flat" GNN architecture, where each MP layer progressively combines the feature of one node with neighbors that are farther and farther away on the graph. **(b)** Example of a "hierarchical" GNN architecture that alternates MP with pooling layers.

## 2.2 Graph Neural Networks

Let $\mathcal{G} = (\mathcal{V}, \mathcal{E})$ be a graph with node features $\mathbf{X}^0 \in \mathbb{R}^{N \times F}$, where $|\mathcal{V}| = N$, and $|\mathcal{E}| = E$. Each row $\mathbf{x}_i^0 \in \mathbb{R}^F$ of the matrix $\mathbf{X}^0$ represents the initial node feature of the node $i$, $\forall i \in \{1, \ldots, N\}$. Through the Message Passing (MP) layers, a GNN implements a local computational mechanism to process graphs (Gilmer et al., 2017). Specifically, each feature vector $\mathbf{x}_v$ is updated by combining the features of the neighboring nodes. After $l$ iterations, $\mathbf{x}_v^l$ embeds both the structural information and the content of the nodes in the $l$–hop neighborhood of $v$. With enough iterations, the feature vectors can be used to classify the nodes or the entire graph. More rigorously, the output of the $l$-th layer of a MP-GNN is:

$$\boldsymbol{x}_v^l = \texttt{COMB}^{(l)}\left(\boldsymbol{x}_v^{l-1}, \texttt{AGGR}^{(l)}(\{\boldsymbol{x}_u^{l-1},\, u \in \mathcal{N}[v]\})\right) \tag{6}$$

where $\texttt{AGGR}^{(l)}$ is a function that aggregates the node features from the neighborhood $\mathcal{N}[v]$ at the $(l-1)$–th iteration, and $\texttt{COMB}^{(l)}$ combines its own features with those of the neighbors.

Traditional GNN architectures are "flat" and consist of a stack of MP layers followed by a final readout (Baek et al., 2021). For graph-level tasks, e.g., graph classification and regression, the readout includes a global pooling operation that combines all the node features at once by taking their sum or average. Such an aggressive pooling operation often fails to extract the global graph properties necessary for the downstream task. On the other hand, GNN architectures that alternate MP with graph pooling layers can gradually distill information into "hierarchical" graph representations. An illustration of a flat and hierarchical GNN architecture is reported in Figure 4. Hierarchical architectures offer several advantages, including the reduction of complexity in the MP operations occurring after pooling (Jin et al., 2021). In addition, by coarsening the graph, graph pooling quickly extends the receptive field of the MP operation, enabling exchanges with distant nodes using fewer MP layers.

## 2.3 Graph pooling

Graph pooling allows building hierarchical GNNs for tasks such as graph classification (Khasahmadi et al., 2020), graph properties prediction (Xu et al., 2024; Leenhouts et al., 2025), node classification (Gao & Ji, 2019; Ma et al., 2020), node clustering (Hansen et al., 2025), graph matching (Liu et al., 2021), physics simulations (Lino et al., 2022), generation with graph diffusion (Valencia et al., 2025), and spatio-temporal forcasting (Cini et al., 2024; Marisca et al., 2024). Existing graph pooling methods can be broadly described through Select-Reduce-Connect (SRC), which provides a general framework to describe different graph pooling operators (Grattarola et al., 2022). According to SRC, a pooling operator, denoted as $\texttt{POOL} : (\boldsymbol{A}, \boldsymbol{X}) \rightarrow (\boldsymbol{A}_{\text{pool}}, \boldsymbol{X}_{\text{pool}})$, is decomposed into three sub-operations:

- Select (`SEL`): maps the original nodes of the graph to a reduced set of nodes, called *supernodes*. The mapping can be represented by a selection matrix $\boldsymbol{S} \in \mathbb{R}^{N \times K}$, where $N$ and $K$ are the number of nodes and supernodes, respectively.

- Reduce (`RED`): generates the features $\boldsymbol{X}_{\mathrm{pool}} \in \mathbb{R}^{K \times F}$ of the supernodes based on the selection matrix and the original node features. Usually, `RED` is implemented as $\boldsymbol{X}_{\mathrm{pool}} = \boldsymbol{S}^\top \boldsymbol{X}$.

- Connect (`CON`): constructs the new adjacency matrix $\boldsymbol{A}_{\mathrm{pool}} \in \mathbb{R}_{\geq 0}^{K \times K}$ based on the selection matrix and the original topology. A streamlined implementation of `CON` is $\boldsymbol{A}_{\mathrm{pool}} = \boldsymbol{S}^\top \boldsymbol{A} \boldsymbol{S}$.

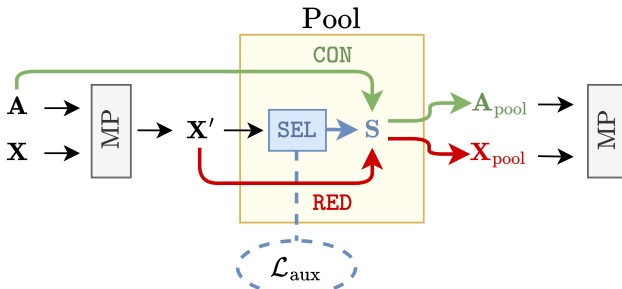

Figure 5: Schematic depiction of a graph pooling layer. The `SEL` operation defines the formation of the supernodes by computing the assignment matrix $\boldsymbol{S}$ from the node embeddings $\boldsymbol{X}'$. The `RED` and `CON` output the pooled node features $\boldsymbol{X}_{\mathrm{pool}}$ and the coarsened adjacency matrix $\boldsymbol{A}_{\mathrm{pool}}$, respectively. Some pooling operators leverage one or more auxiliary losses, $\mathcal{L}_{\mathrm{aux}}$, to influence the formation of the selection matrix $\boldsymbol{S}$ (and, potentially, other components of the GNN).

Figure 5 reports a schematic depiction of a pooling layer showing the interaction of the different components. Different pooling methods are obtained by a specific implementation of these operators and can be broadly categorized into three main families: *score-based*, *one-every-K*, and *soft-clustering* methods.

**Score-Based** methods compute a score for each node using a trainable function in their `SEL` operator. Nodes with the highest scores become the supernodes of the pooled graph. Representatives such as Top-$k$ Pooling (Top-$k$) (Gao & Ji, 2019; Knyazev et al., 2019), ASAPool (Ranjan et al., 2020), SAGPool (Lee et al., 2019), PanPool (Ma et al., 2020), TAPool (Gao et al., 2021), CGIPool (Pang et al., 2021), and IPool (Gao et al., 2022) primarily differ in how they compute the scores or in the auxiliary tasks they optimize to improve the quality of the pooled graph. These methods are computationally efficient and can dynamically adapt the size of the pooled graph, e.g., $K_i = \kappa N_i$, where $\kappa$ is the pooling ratio and $N_i$ and $K_i$ are the sizes of the $i$-th graph before and after pooling, respectively. Score-based methods tend to retain neighboring nodes that have similar features. As a result, entire parts of the graph are under-represented after pooling, reducing the performance in tasks where all the graph structure should be preserved.

**One-Every-$K$** methods pool the graph by uniformly subsampling nodes, extending the concept of one-every-$K$ to irregular graph structures. They are typically efficient and perform pooling inspired by graph-theoretical objectives, such as spectral clustering (Dhillon et al., 2007), maxcut (Bianchi et al., 2020b), and maximal independent sets (Bacciu et al., 2023). Some of these methods lack flexibility because their `SEL` operator neither accounts for node or edge features nor can be influenced by the downstream task's loss. Even if they can adapt the size of the pooled graph $K_i$ to the original graph size $N_i$, the pooling ratio $\kappa$ is determined by the graph-theoretical objective and cannot be specified explicitly.

**Soft-Clustering** methods use `SEL` operators that compute a soft-clustering matrix $\boldsymbol{S}$, which assigns each node to multiple supernodes with different memberships. Representatives such as Diffpool (Ying et al., 2018), MinCut Pool (MinCut) (Bianchi et al., 2020a), and Structpool (Yuan & Ji, 2020), leverage flexible trainable functions guided by auxiliary losses to compute the soft assignments from the node features. As illustrated in Figure 5, the auxiliary loss influences the formation of the selection matrix $\boldsymbol{S}$ (and, potentially, other

parameters of the GNN architecture), ensuring that the partition is consistent with the graph topology and that the clusters are well formed, e.g., that the assignments are sharp and the clusters are balanced. Computing these auxiliary losses typically requires $\mathcal{O}(N^2)$ operations because they require a dense representation of the input adjacency matrix. The quadratic computational cost is generally acceptable in most graph-level tasks, such as graph classification and graph properties prediction, where the size of the graphs ranges from hundreds to a few thousand nodes.

While soft-clustering methods generally achieve high performance due to their flexibility and ability to retain information from the entire graph, they face a primary limitation: they require to predefine the size $K$ of *every* pooled graph, which is fixed for each graph $i$ regardless of its size $N_i$. A typical choice is to set $K = \kappa \bar{N}$, where $\bar{N}$ is the average size of all the graphs in the dataset. Clearly, this might not work well in datasets where the graphs' size varies too much, especially if there are graphs where $N_i < \kappa \bar{N}$. In those cases, the pooling operator *expands* the graph rather than coarsening it, which goes against the principle of pooling.

## 3 Bayesian Nonparametric Pooling for Graphs

We propose BN-Pool, a novel soft-clustering pooling operator grounded in the Bayesian nonparametric theory. The `SEL` function of BN-Pool addresses the main drawbacks of existing soft-clustering methods by learning, for each graph $i$, a pooled graph with a variable number of supernodes $K_i$.

BN-Pool assumes that the observed graph structure can be explained by a latent partition of its nodes. In other words, it assumes that the input adjacency matrix $\boldsymbol{A}$ is generated by an underlying process where each node belongs to a (hidden) cluster, and the probability of finding an edge depends on these cluster memberships. To avoid fixing the number of clusters in advance, BN-Pool places a DP prior over the cluster assignments, allowing the model to adaptively determine the number of clusters needed for each graph. Once the generative model is defined, the `SEL` operator is implemented by performing Bayesian inference: we estimate the posterior distribution of node-to-cluster assignments given the observed graph and node features. This posterior provides soft membership scores that drive the pooling operation.

To ease the notation, we present the method by considering only a single graph. The pseudo-code in Python for implementing all the main operations in BN-Pool is reported in Appendix A.

### 3.1 Definition of the Generative Process

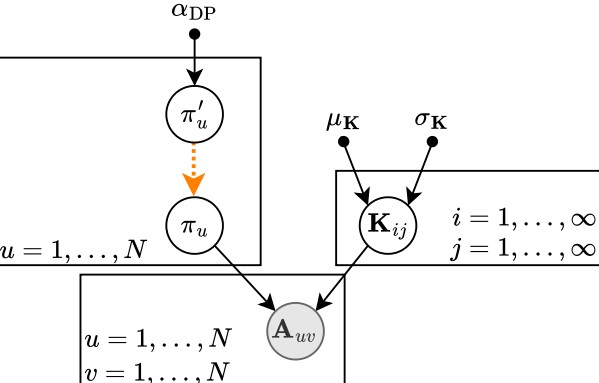

Figure 6: Graphical representation of BN-Pool in plate notation. The orange dotted arrow indicates the stick-breaking construction.

BN-Pool defines a generative process for the adjacency matrix $\boldsymbol{A}$ of the input graph that is similar to the Stochastic Block Model (SBM) (Holland et al., 1983): each node $u$ is associated with a vector $\boldsymbol{\pi}_u$ whose entries indicate the probability that $u$ belongs to a given cluster. The edges are generated according to a block matrix $\boldsymbol{K}$ whose entry $\boldsymbol{K}_{ij}$ represents the unnormalized log-probability of occurrence of an edge between

a node in cluster $i$ and a node in cluster $j$. Unlike in the SBM, we relax the requirement of specifying the number of clusters in advance and leverage the DP to define a prior over an infinite number of clusters. Note that, even if there is an infinite number of clusters, only a few of them are used due to the clustering property of the DP, discussed in Section 2.1.

By exploiting the stick-breaking construction of DPs, we define the generative process of BN-Pool as:

$$
\boldsymbol{K}_{ij} \sim p(\boldsymbol{K}_{ij}) = \begin{cases} \mathcal{N}(\mu_{\boldsymbol{K}}, \sigma_{\boldsymbol{K}}) & \text{if } i = j \\ \mathcal{N}(-\mu_{\boldsymbol{K}}, \sigma_{\boldsymbol{K}}) & \text{if } i \neq j \end{cases}, \qquad \boldsymbol{\pi}'_{ui} \sim p(\boldsymbol{\pi}'_{ui}) = \text{Beta}(1, \alpha_{\text{DP}}),
$$

$$
\boldsymbol{\pi}_{ui} = \boldsymbol{\pi}'_{ui} \prod_{j=1}^{i-1} (1 - \boldsymbol{\pi}'_{uj}), \qquad p_{uv} = \sigma(\boldsymbol{\pi}_u^\top \boldsymbol{K} \boldsymbol{\pi}_v), \qquad \boldsymbol{A}_{uv} \sim p(\boldsymbol{A}_{uv}) = \text{Bernoulli}(p_{uv}),
$$

$$(7)$$

where $u, v \in \mathcal{V}$ are nodes in the input graph, $i, j \in \mathbb{N}$ are cluster indices, and $\sigma(\cdot)$ is the sigmoid function; the hyperparameters $\alpha_{\text{DP}} \in \mathbb{R}^+, \mu_{\boldsymbol{K}} \in \mathbb{R}^+, \sigma_{\boldsymbol{K}} \in \mathbb{R}^+$ define the shape of the prior distributions. The prior distribution on the matrix $\boldsymbol{K}$ defined by $p(\boldsymbol{K}_{ij})$ encodes our assumption that most of the edges link nodes of the same group. The generative process is schematized in Figure 6.

## 3.2 Posterior Estimation

The BNP setting makes the computation of cluster assignments' posterior intractable, and it requires some approximations. We rely on a truncated variational approximation of the posterior (Blei & Jordan, 2004): even if there is an infinite number of clusters, we truncate the posterior by considering a finite value $C$ representing the maximum number of clusters. It is worth highlighting that this does not imply that the model has a fixed number of clusters but, rather, that the model will choose a suitable number of non-empty (i.e., active) clusters $K_i < C$ for the $i$-th graph.

Following the classical mean-field approximation[1], we define two variational distributions: one to model the posterior of the stick fractions $\boldsymbol{\pi}'$, and one to model the posterior of the model parameter $\boldsymbol{K}$. Note that we are interested in the cluster assignment vectors $\boldsymbol{\pi}$, which are fully determined by the stick-breaking construction given the stick fractions $\boldsymbol{\pi}'$. The posterior approximation is expressed as:

$$
q(\boldsymbol{\pi}'_{ui}) = \text{Beta}(\tilde{\boldsymbol{\alpha}}_{ui}, \tilde{\boldsymbol{\beta}}_{ui}), \tag{8}
$$
$$
q(\boldsymbol{K}_{ij}) = \mathcal{N}(\tilde{\boldsymbol{\mu}}_{ij}, \epsilon), \tag{9}
$$

where $\tilde{\boldsymbol{\alpha}}_{ui}, \tilde{\boldsymbol{\beta}}_{ui} \in \mathbb{R}^+, \tilde{\boldsymbol{\mu}}_{ij} \in \mathbb{R}$ for all $u \in \mathcal{V}, i, j \in \{1, \ldots, C\}$ are the variational parameters. The value of $\epsilon$ is fixed a priori, and it is not optimized during the training. While $\tilde{\boldsymbol{\mu}}_{ij}$ are free parameters that we optimize directly, we employ a Multilayer Perceptron (MLP) with parameters $\Theta_{\text{MLP}}$ to estimate $\tilde{\boldsymbol{\alpha}}$ and $\tilde{\boldsymbol{\beta}}$:

$$
\boldsymbol{X}' = \texttt{MP}_{\Theta_{\text{MP}}}(\boldsymbol{X}, \boldsymbol{A}).
$$
$$
\tilde{\boldsymbol{\alpha}}, \tilde{\boldsymbol{\beta}} = \texttt{MLP}_{\Theta_{\text{MLP}}}(\boldsymbol{X}'). \tag{10}
$$

The MLP is applied on the node embeddings $\boldsymbol{X}'$, which are computed by the MP layers with parameters $\Theta_{\text{MP}}$ that are placed in the GNN before the pooling operator. This allows for representing complex relations between hidden and observable variables that usually appear in the posterior distribution by conditioning the posterior on the graph topology, the node (and potentially edge) features, and the downstream task at hand that drives the GNN optimization.

The estimation of variational parameters through a neural network closely resembles the architecture of a Variational Auto-Encoder (VAE). Specifically, the GNN responsible for approximating the posterior acts as the *encoder* in the classical VAE framework, while the SBM serves as the *decoder*, reconstructing the adjacency matrix of the input graph. As discussed in the following, this reconstruction step is central to the training objective, ensuring that the latent node-to-cluster assignments are consistent with the observed structure.

---

[1]The variational distribution is factorized over the latent variables: $p(\boldsymbol{\pi}', \boldsymbol{K} \mid \boldsymbol{A}, \boldsymbol{X}) \approx q(\boldsymbol{\pi}', \boldsymbol{K}) \approx q(\boldsymbol{\pi}')q(\boldsymbol{K})$.

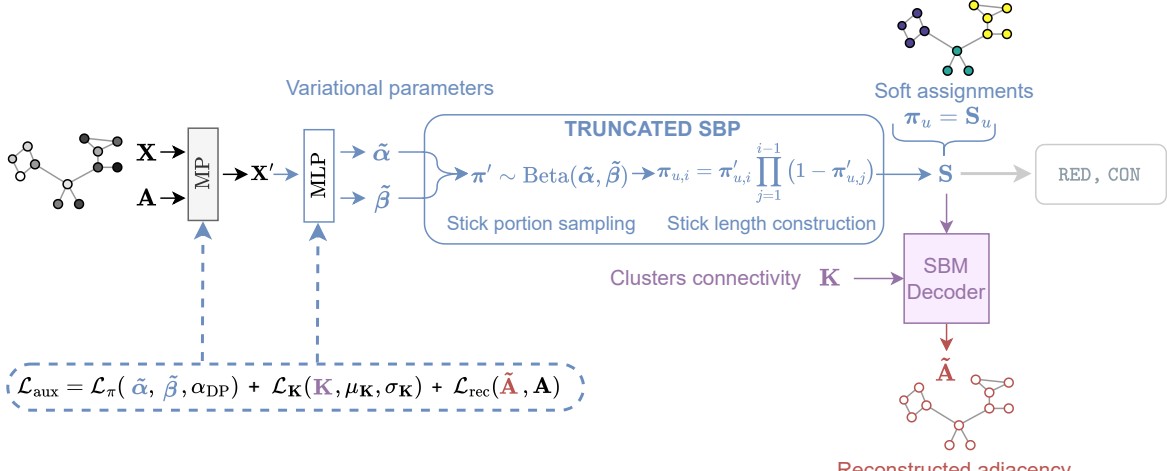

Figure 7: The `SEL` operation of BN-Pool and the components of the auxiliary loss. One or more MP layers (not part of the pooling operator) generate the node embeddings $\boldsymbol{X}'$, which are passed to the MLP that outputs the variational parameters $\tilde{\boldsymbol{\alpha}}$ and $\tilde{\boldsymbol{\beta}}$. These are used by the truncated SBP to generate the assignment matrix $\boldsymbol{S}$, which is passed further to the `RED` and `CON` operators to compute the pooled graph. $\boldsymbol{S}$ is also fed into the SBM decoder for reconstructing the adjacency matrix $\tilde{\boldsymbol{A}}$, conditioned by the block matrix $\boldsymbol{K}$. We note that $\boldsymbol{K}$ and $\tilde{\boldsymbol{A}}$ are necessary solely for the computation of the auxiliary loss $\mathcal{L}_{\mathrm{aux}}$, which influences the parameters of the MLP and all the trainable parameters of previous layers in the GNN.

### 3.3 Graph Pooling Operations

We conclude by casting BN-Pool in the SRC framework. Figure 7 expands Figure 5 by showing details of the `SEL` operation and the auxiliary loss, described in Section 3.4, implemented by BN-Pool. In particular, for each graph $i$, the `SEL` operator generates a cluster assignment matrix $\boldsymbol{S}_i \in \mathbb{R}^{N \times C}$ with $K_i$ columns containing non-zero values. The entry $\boldsymbol{s}_{uj} = \boldsymbol{\pi}_{uj}$ represents the membership of node $u$ to cluster $j$, where we use $\boldsymbol{\pi}_{uj}$ to denote a sample from its posterior $q(\boldsymbol{\pi}_{uj})$ to simplify the notation.

The `RED` and `CON` functions follow the standard implementation of other pooling methods, as described in Section 2.3: $\boldsymbol{X}_{\mathrm{pool}} = \boldsymbol{S}^\top \boldsymbol{X}$ and $\hat{\boldsymbol{A}}_{\mathrm{pool}} = \boldsymbol{S}^\top \boldsymbol{A} \boldsymbol{S}$. In addition, following (Bianchi et al., 2020a), we set the diagonal elements of $\hat{\boldsymbol{A}}_{\mathrm{pool}}$ to zero to prevent self-loops from dominating the propagation in the MP layers after pooling, and we symmetrically normalize it by the nodes' degree: $\boldsymbol{A}_{\mathrm{pool}} = \hat{\boldsymbol{D}}_{\mathrm{pool}}^{-1/2} \hat{\boldsymbol{A}}_{\mathrm{pool}} \hat{\boldsymbol{D}}_{\mathrm{pool}}^{-1/2}$.

### 3.4 Training Procedure

All the neural parameters $\Theta = \{\Theta_{\mathrm{MP}}, \Theta_{\mathrm{MLP}}\}$, and the variational parameters $\tilde{\boldsymbol{\mu}}$, are learned by maximising the Evidence Lower Bound (ELBO):

$$
\log p(\boldsymbol{A}) \geq \underbrace{\sum_u \sum_v \mathbb{E}_{q(\boldsymbol{\pi}')q(\boldsymbol{K})} \left[ \log p(\boldsymbol{A}_{uv} \mid \boldsymbol{\pi}, \boldsymbol{K}) \right]}_{-\mathcal{L}_{\mathrm{rec}}}
$$
$$
\underbrace{- \sum_u \sum_i D_{\mathrm{KL}}(q(\boldsymbol{\pi}'_{ui}) \mid p(\boldsymbol{\pi}'_{ui}))}_{-\mathcal{L}_{\boldsymbol{\pi}}} \quad \underbrace{- \sum_i \sum_j D_{\mathrm{KL}}(q(\boldsymbol{K}_{ij}) \mid p(\boldsymbol{K}_{ij}))}_{-\mathcal{L}_{\boldsymbol{K}}}. \tag{11}
$$

The first term in Equation 11 is the *reconstruction loss* that measures how well the model reconstructs the adjacency matrix. The last two terms measure the distances between the prior and the variational distributions and act as regularisers. While the reconstruction loss $\mathcal{L}_{\mathrm{rec}}$ has a straightforward interpretation,

we can think of $\mathcal{L}_{\boldsymbol{\pi}}$ as the cost of having a certain number of clusters active. Hence, $\mathcal{L}_{\boldsymbol{\pi}}$ reflects the clustering property of the DP in reusing non-empty clusters. On the other hand, $\mathcal{L}_{\boldsymbol{K}}$ penalizes the discrepancy from the connectivity across clusters described by the SBM prior.

In practice, instead of maximising the ELBO in Equation 11, we train the model by minimising the loss:

$$\mathcal{L}_{\text{aux}} = \frac{1}{N^2}\mathcal{L}_{\text{rec}} + \gamma\frac{1}{N^2}\mathcal{L}_{\boldsymbol{\pi}} + \frac{1}{N^2}\mathcal{L}_{\boldsymbol{K}}, \tag{12}$$

where $N$ is the number of nodes in the input graph, and it is used to rescale the losses, while $\gamma$ is a hyperparameter that balances the contrasting effect of $\mathcal{L}_{\text{rec}}$ and $\mathcal{L}_{\boldsymbol{\pi}}$. The interplay between all the loss terms is crucial for an effective adaptive nonparametric method. The normalization and scaling parameters avoid a dominance of the KL divergence and have already been applied on VAEs (Higgins et al., 2017; Asperti & Trentin, 2020). We refer to the loss in Equation 12 as *auxiliary* since, during pooling, it is combined with the supervised loss of the downstream task. It is important to note that these terms are *not* independent auxiliary objectives but are intrinsically linked as they constitute the ELBO. Therefore, removing one of these terms would not be meaningful, as it would invalidate the derivation of the variational lower bound and the model's probabilistic foundation.

The training is performed by employing the Stochastic Gradient Variational Bayes (SGVB) framework (Kingma & Welling, 2014), where the expectation in the reconstruction term is approximated with a Monte Carlo estimate of the binary cross-entropy between the true edges and the probabilities predicted by the model:

$$\mathcal{L}_{\text{rec}} \approx \sum_{t=1}^{T}\sum_{u}\sum_{v} -\boldsymbol{A}_{uv}\log p_{uv}^{t} - (1 - \boldsymbol{A}_{uv})\log(1 - p_{uv}^{t}), \tag{13}$$

where $T$ is the number of samples used for the Monte Carlo approximation, and $p_{uv}^{t} = \sigma(\sum_{i}\sum_{j}\boldsymbol{\pi}_{ui}^{t}\tilde{\boldsymbol{\mu}}_{ij}\boldsymbol{\pi}_{vj}^{t})$, where $\boldsymbol{\pi}_{u}^{t}$ and $\boldsymbol{\pi}_{v}^{t}$ are the $t$-th samples of the soft assignments nodes $u$ and $v$, respectively. Each sampling step $\boldsymbol{\pi}_{ui}' \sim \text{Beta}(\tilde{\boldsymbol{\alpha}}_{ui}, \tilde{\boldsymbol{\beta}}_{ui})$ needed to approximate $\mathcal{L}_{\text{rec}}$ is not differentiable and prevents the gradient from being backpropagated to the neural parameters $\Theta$. A common approach to solve this issue is the reparameterization trick (Kingma & Welling, 2014), which, however, cannot be applied to the Beta distribution (Figurnov et al., 2018). Therefore, in BN-Pool, we implement the backpropagation by approximating the pathwise gradient of the sampled values w.r.t. the distribution parameters[2] (Jankowiak & Obermeyer, 2018).

To reduce the stochasticity of the approximation, we assume that the variational distribution $q(\boldsymbol{K})$ has a low variance (i.e., $\varepsilon \to 0$ in Equation 9) and directly use the variational parameter $\tilde{\boldsymbol{\mu}}$ rather than sampling the cluster connectivity from its variational distribution. Finally, we initialise the neural parameters $\Theta_{\text{MLP}}$ by using the default initialisation of the backend (He et al., 2015), while the variational parameter $\tilde{\boldsymbol{\mu}}$ of the cluster connectivity matrix is initialised by setting the elements on-diagonal (off-diagonal) equal to $\eta_{\boldsymbol{K}}$ $(-\eta_{\boldsymbol{K}})$, where $\eta_{\boldsymbol{K}}$ is a hyperparameter.

### 3.4.1 $\mathcal{L}_{\text{rec}}$ with $O(E)$ complexity.

The computation of $\mathcal{L}_{\text{rec}}$ requires $\mathcal{O}(N^2)$ operations, as the number of samples $T$ is negligible compared to $N^2$. While this complexity is consistent with other soft-clustering pooling methods that rely on a dense representation of the input adjacency matrix (e.g., DiffPool (Ying et al., 2018)), we introduce a sparse variant of BN-Pool that reduces the complexity to $\mathcal{O}(E)$.

The sparse variant of BN-Pool operates only on the observed edges, i.e., the set of node pairs $(u, v)$ such that $\boldsymbol{A}_{u,v} = 1$, and on a sampled set of missing edges, i.e., pairs such that $\boldsymbol{A}_{u,v} = 0$. Concretely, let $\mathcal{E}^{-} = \{(u, v) \mid (u, v) \notin \mathcal{E}\}$ be a sampled set of missing edges such that $|\mathcal{E}^{-}| = E$. We then define a sparse surrogate of the reconstruction loss by restricting the summation in equation 13 to the positive edges and to the sampled negatives:

$$\mathcal{L}_{\text{rec}} \approx -\sum_{t=1}^{T}\left(\sum_{(u,v)\in\mathcal{E}}\log p_{uv}^{t} + \sum_{(u,v)\in\mathcal{E}^{-}}\log(1 - p_{uv}^{t})\right). \tag{14}$$

---

[2]This approximation is already implemented in the PyTorch library. See Appendix A for more details about our implementation.

Note that, differently from equation 13, here the references to the adjacency matrix $\boldsymbol{A}$ are omitted, since $\boldsymbol{A}_{u,v} = 1$ if and only if $(u,v) \in \mathcal{E}$. By construction, the summation in equation 14 contains $2E$ terms. To ensure an overall complexity of $O(E)$, we compute the values $p_{u,v}^t$ only for the node pairs appearing in the summation, avoiding materialization of the full $N \times N$ matrix. All other components of the training procedure remain unchanged, except for the constant used to normalize the terms of $\mathcal{L}_{\text{aux}}$: instead of using $N^2$, we normalize each term by $|\mathcal{E}| + |\mathcal{E}^-|$.

While this surrogate objective is conceptually simple, its implementation is not straightforward and must be done carefully. For example, sampling the set of negative edges $\mathcal{E}^-$ must avoid $O(N^2)$ memory allocation and should be performed efficiently on the GPU to prevent performance degradation. We discuss the implementation details in Appendix A.4.

### 3.5 Prior Hyperparameters Interpretation

To fully define the BN-Pool model, we have to specify three hyperparameters: $\alpha_{\text{DP}}$, $\mu_{\boldsymbol{K}}$, and $\sigma_{\boldsymbol{K}}$. The probabilistic nature of our method allows for a direct interpretation that facilitates their tuning.

The value of $\alpha_{\text{DP}} \in \mathbb{R}^+$ defines the shape of the prior over the cluster assignments; in particular, it specifies the concentration of the DP. To understand the effect of $\alpha_{\text{DP}}$, we recall that the loss $\mathcal{L}_{\boldsymbol{\pi}}$ is the cost to pay to have a certain number of clusters active. The value $\alpha_{\text{DP}}$ is inversely proportional to the price to activate a new cluster: low values force the model to use a few clusters (only one in the extreme case). Conversely, higher values do not penalize the model when it uses more clusters to reduce the reconstruction loss. Since in practice we truncate the posterior to at most $C$ clusters, too high values of $\alpha_{\text{DP}}$ can create degenerate solutions where the last cluster is always used.

The other two hyperparameters $\mu_{\boldsymbol{K}} \in \mathbb{R}^+$ and $\sigma_{\boldsymbol{K}} \in \mathbb{R}^+$ specify the prior over the block matrix $\boldsymbol{K}$, which affects the reconstruction loss. Again, the most intuitive way to understand the effect of $\boldsymbol{K}$ is in terms of costs: if the value $\boldsymbol{K}_{ij}$ is positive (negative), the price of creating an edge between a node in cluster $i$ and a node in cluster $j$ is low (high). Thus, to encode our prior belief that most of the edges appear between nodes in the same cluster, we impose that the elements on the diagonal are positive with value $\mu_{\boldsymbol{K}}$ (i.e., intra-cluster edges are cheap), while the off-diagonal elements are negative with value $-\mu_{\boldsymbol{K}}$ (i.e., inter-cluster edges are costly). The hyperparameter $\sigma_{\boldsymbol{K}}$ controls the strength of the prior: the lower, the more the posterior matches the prior rather than the data.

The values of $\mu_{\boldsymbol{K}}$ and $\sigma_{\boldsymbol{K}}$ also affect the number of active clusters. For example, the degenerate solution that assigns all the nodes to the first cluster satisfies the clusterization property of the DP. However, by referring to Equation 13, this means paying $-\log(1 - \sigma(\tilde{\boldsymbol{\mu}}_{11})) = -\log \sigma(-\tilde{\boldsymbol{\mu}}_{11})$ every time $\boldsymbol{A}_{uv} = 0$. If the posterior matches our prior (i.e., $\tilde{\boldsymbol{\mu}}_{11} \approx \mu_{\boldsymbol{K}}$), this results in a high cost since $\mu_{\boldsymbol{K}} \gg 0$ implies $-\log \sigma(-\mu_{\boldsymbol{K}}) \gg 0$; thus, the model will likely prefer to reduce $\mathcal{L}_{\text{rec}}$ at the price of having more clusters, i.e., a larger $\mathcal{L}_{\boldsymbol{\pi}}$. Finally, we note that while the other hyperparameters (truncation level $C$, number of samples $T$, and initialization of the neural and variational parameters $\Theta$ and $\eta_{\boldsymbol{K}}$) influence the training procedure, they do not affect the model definition.

## 4 Related work

BN-Pool belongs to the family of Soft-Clustering pooling methods discussed in Section 2.3 and the closest approach is Diffpool (Ying et al., 2018), which employs an auxiliary loss $\|\boldsymbol{A} - \boldsymbol{S}\boldsymbol{S}^\top\|_F$ to align the assignments to the graph topology. In this work, we go beyond the formulation of such a simple loss and define a whole generative process for the adjacency matrix.

Related to our work is the Dirichlet Graph Variational Auto-Encoder (DGVAE) (Li et al., 2020), which defines a VAE with a Dirichlet prior over the latent variables to cluster graph nodes. We extend DGVAE in different ways. First, we define a more flexible generative process for the adjacency matrix thanks to the block matrix $\boldsymbol{K}$. Second, we allow an infinite number of clusters by specifying a DP prior over the latent variables. Finally, we do not rely on the Laplace approximation of the Dirichlet distribution, whose behavior is similar to a Gaussian prior (Joo et al., 2020).

The Stick-Breaking Variational Auto-Encoder (SB-VAE) (Nalisnick & Smyth, 2017) shares our idea of specifying a nonparametric prior over the hidden variables by using a DP prior that leverages the stick-breaking construction, but does not focus on graphs. We also employ a different approximation of the posterior, which is based on pathwise gradients rather than the Kumaraswamy distribution (Kumaraswamy, 1980).

Another work that shares similarities with our method is (Mehta et al., 2019). It introduces a sparse VAE for overlapping SBM that also allows an infinite number of clusters, but uses a different nonparametric prior: the Indian Buffet Process (IBP) (Griffiths & Ghahramani, 2011). The IBP is suitable to model multiple cluster membership, i.e., a node can belong to more than one cluster, which is not desirable in the context of pooling. Moreover, Mehta et al. use for each node another dense latent variable with a Gaussian prior to gain more flexibility during the generation process of the adjacency matrix. Instead, in BN-Pool all the information useful for the generation is encoded in the soft cluster assignments $\boldsymbol{S}$.

## 5 Experiments

The purpose of our experiments is twofold. Since BN-Pool is the first BNP pooling method, we start by analyzing its ability to detect communities on a single graph. Then, we test the effectiveness of BN-Pool in GNNs for graph-level tasks such as graph classification and graph regression. In all experiments, we use very simple GNN models to better quantify the differences in performance between each pooling method. Indeed, while GNNs with larger capacity can achieve SOTA performance, it is harder to disentangle the actual contribution of the pooling operator in a more complex model.

In the following, we consider the configurations of the hyperparameters of BN-Pool specified in Table 1. As discussed in Section 3.5, the value of each parameter can be set according to the characteristics of the dataset at hand or by monitoring some performance metrics while training. In each experiment, we select the configuration that yields the lowest value of the reconstruction loss $\mathcal{L}_{\text{rec}}$ in the node clustering task and the highest validation accuracy in the graph classification task.

Table 1: Values of the hyperparameters of BN-Pool considered.

| Hyperparameter | Values |
|---|---|
| $\alpha_{\text{DP}}$ | 0.1, 1.0, 10.0, 30.0 |
| $\mu_{\boldsymbol{K}}$ | 0.1, 1.0, 10.0, 30.0 |
| $\sigma_{\boldsymbol{K}}$ | 0.1, 1.0 |

We found that setting $\alpha_{\text{DP}} = 10.0$, $\mu_{\boldsymbol{K}} = 1.0$, and $\sigma_{\boldsymbol{K}} = 1.0$ yields generally good performance and, thus, it represents our default configuration. Regarding the other hyperparameters, we kept the truncation level $C = 50$, the number of particles $T = 1$, and the initialization of the variational parameter $\eta_{\boldsymbol{K}} = 1.0$ fixed in all experiments.

The code to reproduce all the experiments presented in this paper is publicly available [3]. In addition, BN-Pool is available on the Torch Geometric Pool library (Bianchi et al., 2025)[4].

### 5.1 Community detection

This task consists of learning a partition of the graph nodes in an unsupervised fashion, only based on the node features and the graph topology. Even if our primary focus is on graph pooling, this experiment allows us to evaluate the auxiliary losses in terms of the consistency between the node labels $\boldsymbol{y}$ and the cluster assignments.

The architecture used for clustering consists of a stack of MP layers that generate the feature vectors $\boldsymbol{X'}$. As MP layers we used two Graph Convolutional Network (GCN) layers (Kipf & Welling, 2017) with 32 hidden

---

[3]https://github.com/NGMLGroup/Bayesian-Nonparametric-Graph-Pooling
[4]https://github.com/tgp-team/torch-geometric-pool

units and ELU activations (Clevert, 2015). The features $\boldsymbol{X}'$ are processed by the SEL operator that produces the cluster assignments $\boldsymbol{S}$. Since clustering is an unsupervised task, the GNN is trained by minimizing only the auxiliary losses. The architecture used for clustering is depicted in Figure 8.

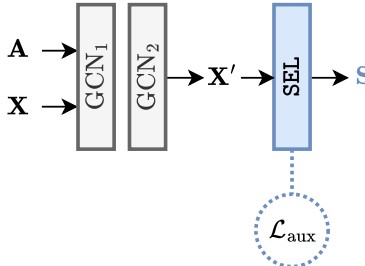

Figure 8: Architecture used for node clustering task.

Before training, we apply to the adjacency matrix the same pre-transform used in Just-balance Graph Neural Network (JBGNN):

$$\boldsymbol{A} \rightarrow \boldsymbol{I} - \delta * \boldsymbol{L}, \tag{15}$$

where $\boldsymbol{L}$ is the symmetrically normalized graph Laplacian and $\delta$ is a constant that we set to 0.85 as in (Bianchi, 2023). As the training algorithm, we used Adam (Kingma & Ba, 2015) with initial learning rate $1e-3$. For BN-Pool, we increased $\gamma$ defined in Equation 12 from 0 to 1 over the first $5,000$ epochs according to a cosine scheduler. This scheduling procedure, often referred to as "KL annealing" (Bowman et al., 2016; Sønderby et al., 2016), is a standard practice when training VAEs to prevent the model from converging to degenerate solutions early in the training due to, e.g., posterior collapse.

During training, we monitored the auxiliary losses for early stopping with patience $1,000$. When the GNN was configured with BN-Pool, we monitored only $\mathcal{L}_{\text{rec}}$ since $\mathcal{L}_{\boldsymbol{K}}$ and $\mathcal{L}_{\boldsymbol{\pi}}$ are regularization losses that usually increase and might dominate the total loss.

Clustering performance is commonly measured with Normalized Mutual Information (NMI), Completeness, and Homogeneity scores, which only work with hard cluster assignments. While the latter can be obtained by taking the argmax of a soft assignment, the discretization process can discard useful information. Consider, for example, a case where two nodes $u$ and $v$ have assignment vectors $\boldsymbol{s}_u = [0, 0.5, 0.5, 0]$ and $\boldsymbol{s}_v = [0, 0.5, 0, 0.5]$. Taking the argmax would map both nodes in the 2nd cluster, even if the two assignment vectors are clearly distinguishable. This problem is exacerbated when we do not fix the number of clusters $K$ equal to the true number of classes; in this case, there is no direct correspondence between the clusters and the classes, and nothing prevents different classes from being represented by partially overlapping assignment vectors with multiple non-zero entries.

Therefore, to measure the agreement between $\boldsymbol{S}$ and $\boldsymbol{y}$, we first consider the cosine similarity between the cluster assignments and the one-hot representation of the node labels:

$$\text{COS} = \frac{\sum_{i,j} \left[ \boldsymbol{S}\boldsymbol{S}^\top \odot \boldsymbol{Y}\boldsymbol{Y}^\top \right]_{i,j}}{\sqrt{\sum_{i,j} \left[ \boldsymbol{S}\boldsymbol{S}^\top \right]_{i,j} + \sum_{i,j} \left[ \boldsymbol{Y}\boldsymbol{Y}^\top \right]_{i,j}}} \tag{16}$$

where $\boldsymbol{Y} = \texttt{one-hot}(\boldsymbol{y})$. As a second measure, we consider the accuracy (ACC) obtained by training a simple logistic regression classifier to predict $\boldsymbol{y}$ from $\boldsymbol{S}$.

We compare the performance of BN-Pool with the assignments obtained by four other soft-clustering pooling methods, DiffPool (Ying et al., 2018), MinCut (Bianchi et al., 2020a), Deep Modularity Network (DMoN) (Tsitsulin et al., 2023), and JBGNN (Bianchi, 2023), which are optimized by minimizing their own auxiliary losses. Importantly, we note that the other methods leverage supervised information by setting the number of clusters $K$ equal to the number of node classes, while BN-Pool is completely unsupervised.

Table 2: Mean and standard deviations of ACC and COS for vertex clustering.

| Method | Community | | Cora | | CiteSeer | | PubMed | | DBLP | |
|---|---|---|---|---|---|---|---|---|---|---|
| | ACC | COS | ACC | COS | ACC | COS | ACC | COS | ACC | COS |
| DiffPool | $81.9_{\pm1.3}$ | $62.9_{\pm0.6}$ | $50.4_{\pm1.1}$ | $43.3_{\pm0.0}$ | $37.9_{\pm1.4}$ | $\mathbf{42.4}_{\pm0.0}$ | $52.4_{\pm0.7}$ | $59.8_{\pm0.0}$ | $49.5_{\pm4.9}$ | $57.4_{\pm0.0}$ |
| MinCut | $97.1_{\pm0.3}$ | $\mathbf{94.3}_{\pm0.5}$ | $57.0_{\pm2.1}$ | $40.1_{\pm1.8}$ | $\mathbf{54.3}_{\pm5.0}$ | $36.9_{\pm3.8}$ | $61.3_{\pm0.2}$ | $46.6_{\pm0.3}$ | $69.2_{\pm3.4}$ | $52.5_{\pm3.9}$ |
| DMoN | $96.2_{\pm0.9}$ | $92.5_{\pm1.6}$ | $57.9_{\pm3.8}$ | $40.1_{\pm2.3}$ | $50.7_{\pm2.4}$ | $34.6_{\pm1.6}$ | $59.6_{\pm1.4}$ | $45.5_{\pm0.7}$ | $63.7_{\pm3.2}$ | $45.4_{\pm1.3}$ |
| JBGNN | $83.9_{\pm8.7}$ | $83.0_{\pm8.9}$ | $55.4_{\pm2.4}$ | $39.0_{\pm2.8}$ | $48.1_{\pm5.0}$ | $36.1_{\pm3.3}$ | $55.8_{\pm3.8}$ | $44.6_{\pm2.0}$ | $68.6_{\pm1.8}$ | $53.0_{\pm4.4}$ |
| BN-Pool | $\mathbf{98.5}_{\pm0.5}$ | $83.0_{\pm1.4}$ | $\mathbf{66.8}_{\pm1.0}$ | $\mathbf{47.7}_{\pm1.3}$ | $47.9_{\pm1.7}$ | $37.8_{\pm0.3}$ | $\mathbf{81.3}_{\pm0.5}$ | $\mathbf{62.5}_{\pm0.7}$ | $\mathbf{75.2}_{\pm0.7}$ | $\mathbf{58.5}_{\pm0.7}$ |

As datasets, we consider *Community*, a synthetic dataset generated from a SBM, and four real-world citation networks. The details about the datasets are in Appendix B. Table 2 reports the results and shows that, despite not knowing the real number of classes, BN-Pool achieves excellent performance.

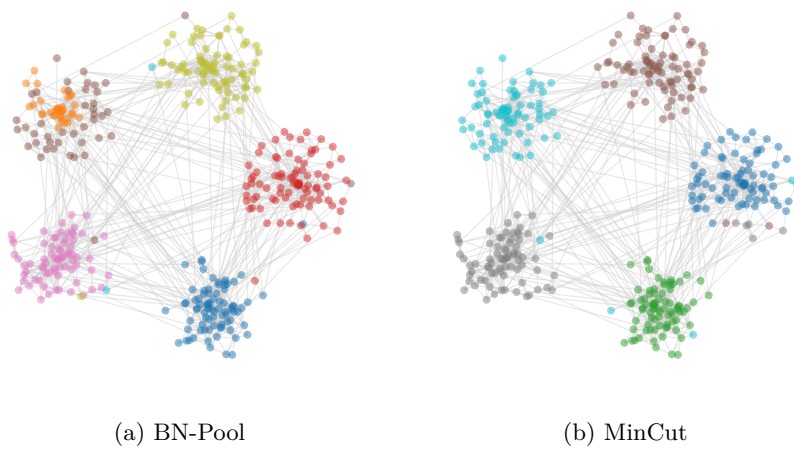

(a) BN-Pool
(b) MinCut

Figure 9: Clusters found on a 5-community graph.

**Discussion.** Figure 9a shows a typical situation where BN-Pool splits a community in two. This happens if there are a few edges within the community, and increasing $K$ yields more compact clusters. This cannot occur in other soft-clustering methods such as MinCut. The latter always finds the same predefined number of clusters ($K = 5$ in this case, see Figure 9b) but creates more spurious clusters.

Figure 10 shows the original adjacency matrix of the Cora dataset, a visualization of the class labels ($\boldsymbol{YY}^\top$), and the adjacency matrix reconstructions $\boldsymbol{SKS}^\top$ and $\boldsymbol{SS}^\top$ obtained by BN-Pool and MinCut, respectively. While the $\boldsymbol{SKS}^\top$ produced by BN-Pool follows more closely the actual sparsity pattern of the adjacency matrix, in MinCut $\boldsymbol{SS}^\top$ has a block structure.

This difference is explained by the different optimization objectives: while BN-Pool tries to reconstruct the whole adjacency matrix, MinCut recovers the communities by cutting the smallest number of edges. In addition, MinCut uses a regularization to encourage clusters to have the same size. This makes it difficult to isolate the two smallest clusters that, instead, are distinguishable in BN-Pool. Given that in Cora the average edge density between nodes of the same class is only 0.0065, a natural way for BN-Pool to lower $\mathcal{L}_{\text{rec}}$ is to activate new clusters and generate assignments with multiple non-zero, yet low, membership values.

To better visualize this behavior, in Figure 11, we show the cluster assignments $\boldsymbol{S}$, split according to the node classes, found by BN-Pool on Cora. We see that there is no direct correspondence between the classes and the clusters, since each class is assigned to multiple clusters. This is expected when we do not fix the number of clusters equal to the number of classes, like in the case of BN-Pool that, potentially, can activate an infinite number of clusters. We also notice that the same clusters are active across different classes, albeit with different membership values. Despite such an overlap, there is a clear and consistent pattern in terms

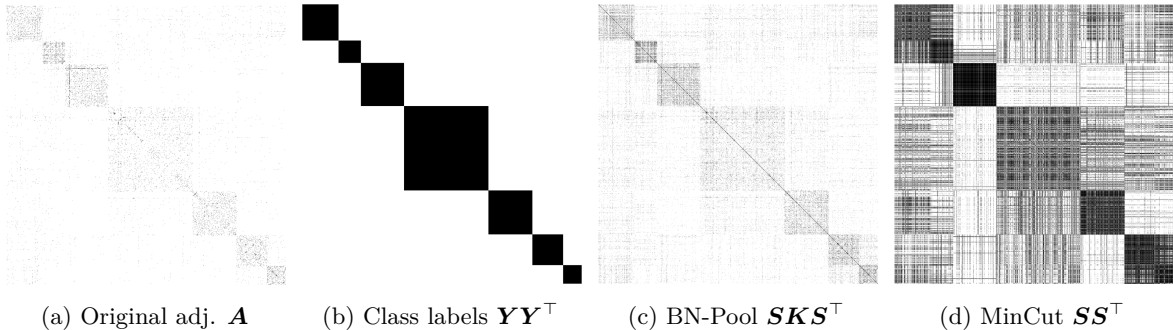

(a) Original adj. $\boldsymbol{A}$     (b) Class labels $\boldsymbol{Y}\boldsymbol{Y}^\top$     (c) BN-Pool $\boldsymbol{S}\boldsymbol{K}\boldsymbol{S}^\top$     (d) MinCut $\boldsymbol{S}\boldsymbol{S}^\top$

Figure 10: Adjacency matrix of Cora, class labels visualization, and adjacency matrix reconstruction by BN-Pool and MinCut.

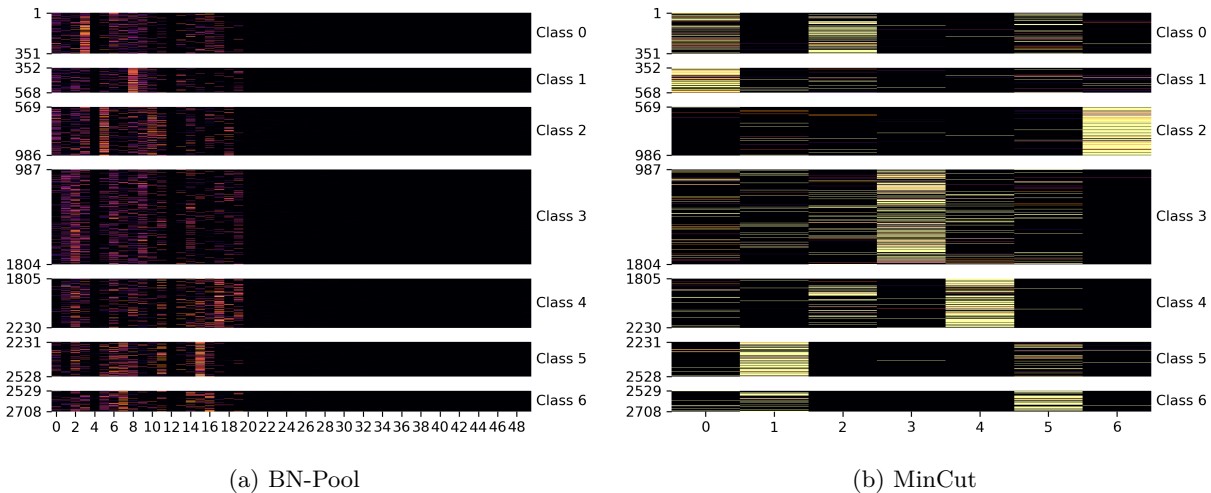

(a) BN-Pool          (b) MinCut

Figure 11: Cluster assignments $\boldsymbol{S}$ found on Cora.

of cluster memberships for each class. It is important to notice that the membership values are lower for the nodes of class 3, which is the most populated in the graph. As discussed in Section 5.1, activating many clusters with low membership values is a natural solution found by BN-Pool to reduce $\mathcal{L}_{\text{rec}}$ when the intra-class density is very low, like in Cora (0.0065).

The clusters found by MinCut on Cora are very different, as shown in Figure 11b. MinCut relies on supervision to set the number of clusters equal to the number of class labels. While this provides a good correspondence between the classes and the clusters, it limits the extent to which MinCut can split a class into multiple clusters, encoding nodes of the same class differently. This implies that if there is a significant variability within each class, MinCut might assign only some of its nodes to the right cluster.

## 5.2 Graph-level tasks

In graph classification and regression, a target $y_i$ is assigned to the $i$-th graph $(\boldsymbol{A}_i, \boldsymbol{X}_i)$. Unlike in the community detection task, here the GNN is optimized by jointly minimizing the task loss (e.g., cross-entropy or MSE) between the true and predicted targets and the auxiliary loss $\mathcal{L}_{\text{aux}}$. The architecture used for graph classification and regression is depicted in Figure 12.

Before and after pooling, we use a Graph Isomorphism Network (GIN) (Xu et al., 2019) layer with 32 hidden units and ELU activations. The readout is an MLP with $[32 \times 32 \times 16 \times N_{\text{class}}]$ units, dropout 0.5, and ELU activation. For datasets with edge features, we replace the first GIN layer with the MP operator proposed

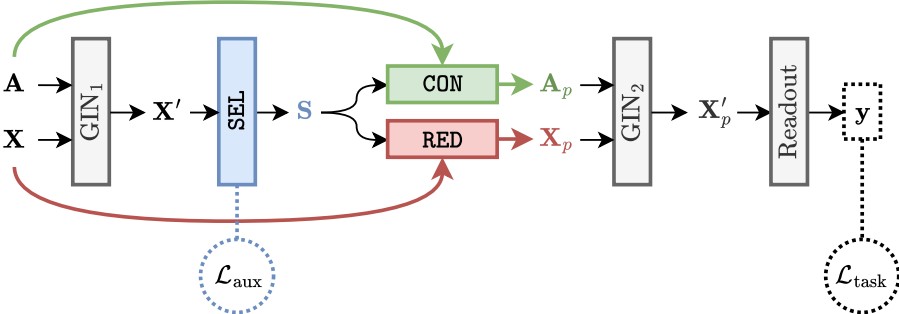

Figure 12: Architecture for graph classification and regression.

by (Hu et al., 2019). For the MolHiv and Peptides-struct datasets, we use a slightly modified architecture with two MP layers before pooling and two after pooling, each with 64 hidden units. After each MP layer, we inserted a dropout layer with probability 0.1 and a batch normalization layer. In addition, we use the standard `AtomEncoder` and `BondEncoder` provided by the OGB library[5] with embedding dimension 100 to transform the original node and edge features. Like in the node clustering setting, we apply the pre-transform in Equation 15. In those datasets containing edge features, we assign to the self-loops that we introduce zero vectors as surrogate features.

While BN-Pool can autonomously discover the number of nodes $K_i$ of each pooled graph, we need to specify the size of the pooled graphs $K$ for the other Soft-Clustering pooling methods and the pooling ratio $\kappa$ for the Score-Based methods. Therefore, for every dataset, we set $\kappa = 0.5$ and $K = 0.5\bar{N}$, where $\bar{N}$ represents the average number of nodes in a given dataset. We note that this is the standard rule of thumb used in the original papers proposing the competing methods. In Appendix C we report the results obtained by the Soft-Clustering methods for different values of $K$.

As an optimizer, we used Adam with an initial learning rate $5e - 4$. Regarding the callbacks, we monitored the validation accuracy and lowered the learning rate by a factor of 0.5 after a plateau of 30 epochs and performed early stopping with patience 100 epochs. For BN-Pool, we increased $\gamma$ from 0 to 1 over the first 50 epochs using a cosine scheduler.

In addition to the poolers from Section 5.1, here we also compare against two additional Soft-Clustering operators, Higher-Order Clustering and Pooling (HOSC) (Duval & Malliaros, 2022) and EigenPooling (Eigen) (Ma et al., 2019), and Score-Based and One-Every-$K$ pooling operators, such as Top-$k$ (Gao & Ji, 2019; Knyazev et al., 2019), Edge-Contraction Pooling (ECPool) (Diehl, 2019), $k$ Maximal Independent Sets Pooling ($k$-MIS) (Bacciu et al., 2023), Graclus (Defferrard et al., 2016), and Structural Entropy Pooling (SEP) (Wu et al., 2022), which have no auxiliary losses. We also consider a baseline without hierarchical pooling, consisting only of a stack of MP layers. As datasets, we consider TUDataset benchmark (Morris et al., 2020), including Colors3 (Knyazev et al., 2019), GCB-H (Bianchi et al., 2022), ogbg-molhiv (Wu et al., 2018), Peptides-struct (Dwivedi et al., 2022), and Multipartite (Abate & Bianchi, 2025). The details about the datasets are in Appendix B.

**Discussion.** We report the results in Table 3 and Table 12. In general, BN-Pool performs on par or better than any other pooling operator, especially those from the Soft-Clustering family. This indicates that BN-Pool can effectively (i) find a meaningful number of clusters and (ii) learn more compact pooled graphs without sacrificing useful information. Noteworthy results are obtained on the datasets Colors-3 and Enzymes, where BN-Pool significantly outperforms any other pooling method and sets the new SOTA.

Figure 13 shows the actual node features from a graph from the GCB-H dataset and the node-to-supernode assignments found by different pooling operators. Interestingly, BN-Pool creates clusters that match the node features well (Figure 13b). By contrast, MinCut, which is also a Soft-Clustering method, places nodes with

---

[5]https://github.com/snap-stanford/ogb

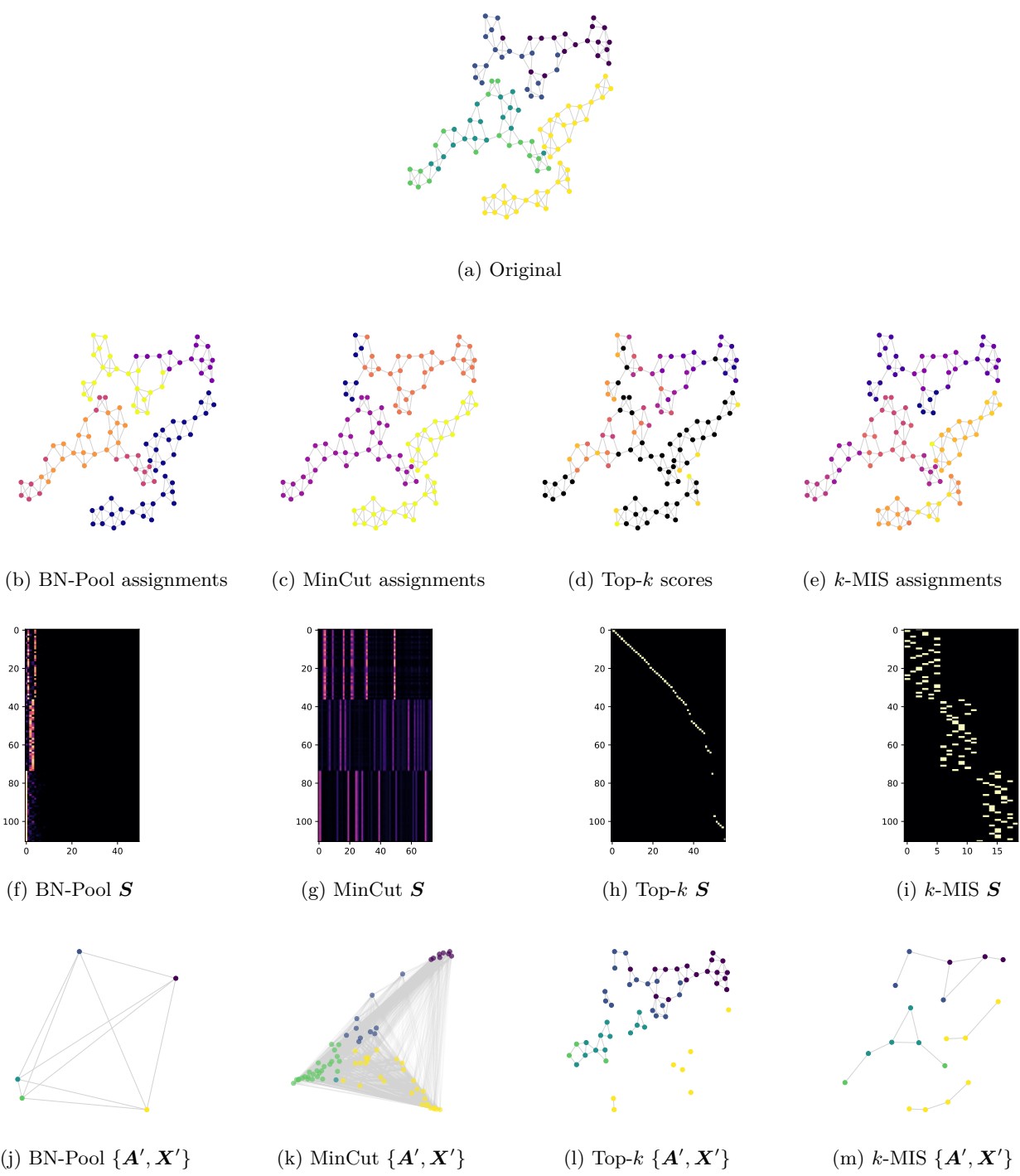

(a) Original

(b) BN-Pool assignments  (c) MinCut assignments  (d) Top-$k$ scores  (e) $k$-MIS assignments

(f) BN-Pool $\boldsymbol{S}$  (g) MinCut $\boldsymbol{S}$  (h) Top-$k$ $\boldsymbol{S}$  (i) $k$-MIS $\boldsymbol{S}$

(j) BN-Pool $\{\boldsymbol{A}', \boldsymbol{X}'\}$  (k) MinCut $\{\boldsymbol{A}', \boldsymbol{X}'\}$  (l) Top-$k$ $\{\boldsymbol{A}', \boldsymbol{X}'\}$  (m) $k$-MIS $\{\boldsymbol{A}', \boldsymbol{X}'\}$

Figure 13: Example from GCB-H.

Table 3: Mean and standard deviations of the graph classification accuracy (ROC-AUC for MolHiv and MAE for Pep-struct).

| Pooler | Collab | Colors3 | Mutagenicity | NCI1 | RedditB | MUTAG | Enzymes | Proteins | MolHiv | Pep-struct | Multip. |
|---|---|---|---|---|---|---|---|---|---|---|---|
| – | $70_{\pm4}$ | $74_{\pm9}$ | $78_{\pm1}$ | $73_{\pm3}$ | $86_{\pm1}$ | $78_{\pm13}$ | $33_{\pm13}$ | $71_{\pm4}$ | $74_{\pm2}$ | $.295_{\pm.007}$ | $14_{\pm12}$ |
| Graclus | $72_{\pm3}$ | $68_{\pm1}$ | $80_{\pm2}$ | $77_{\pm2}$ | $90_{\pm3}$ | $82_{\pm12}$ | $33_{\pm7}$ | $73_{\pm4}$ | $74_{\pm3}$ | $.264_{\pm.001}$ | $48_{\pm2}$ |
| ECPool | $72_{\pm3}$ | $69_{\pm2}$ | $80_{\pm2}$ | $77_{\pm3}$ | $\mathbf{91}_{\pm2}$ | $84_{\pm12}$ | $35_{\pm8}$ | $74_{\pm5}$ | $74_{\pm1}$ | $.262_{\pm.006}$ | $51_{\pm2}$ |
| $k$-MIS | $71_{\pm2}$ | $84_{\pm1}$ | $79_{\pm2}$ | $75_{\pm3}$ | $90_{\pm2}$ | $83_{\pm10}$ | $33_{\pm8}$ | $73_{\pm5}$ | $74_{\pm2}$ | $.263_{\pm.001}$ | $\mathbf{63}_{\pm2}$ |
| Top-$k$ | $72_{\pm2}$ | $78_{\pm1}$ | $75_{\pm3}$ | $73_{\pm2}$ | $77_{\pm2}$ | $82_{\pm10}$ | $29_{\pm7}$ | $74_{\pm5}$ | $76_{\pm1}$ | $.266_{\pm.000}$ | $45_{\pm3}$ |
| SEP | $72_{\pm3}$ | $71_{\pm1}$ | $80_{\pm2}$ | $77_{\pm3}$ | $90_{\pm1}$ | $81_{\pm9}$ | $40_{\pm6}$ | $73_{\pm7}$ | $75_{\pm0}$ | $.350_{\pm.002}$ | $61_{\pm2}$ |
| DiffPool | $70_{\pm2}$ | $65_{\pm1}$ | $78_{\pm2}$ | $75_{\pm2}$ | $90_{\pm2}$ | $81_{\pm11}$ | $36_{\pm7}$ | $75_{\pm3}$ | $70_{\pm4}$ | $.276_{\pm.018}$ | $56_{\pm3}$ |
| MinCut | $70_{\pm2}$ | $69_{\pm1}$ | $78_{\pm3}$ | $73_{\pm3}$ | $87_{\pm2}$ | $81_{\pm12}$ | $34_{\pm9}$ | $\mathbf{77}_{\pm5}$ | $76_{\pm1}$ | $.265_{\pm.003}$ | $56_{\pm3}$ |
| DMoN | $68_{\pm2}$ | $69_{\pm2}$ | $80_{\pm2}$ | $73_{\pm3}$ | $88_{\pm2}$ | $82_{\pm11}$ | $37_{\pm7}$ | $76_{\pm4}$ | $\mathbf{77}_{\pm1}$ | $.280_{\pm.001}$ | $62_{\pm3}$ |
| JBGNN | $72_{\pm2}$ | $68_{\pm2}$ | $80_{\pm2}$ | $78_{\pm3}$ | $90_{\pm1}$ | $87_{\pm14}$ | $39_{\pm6}$ | $75_{\pm5}$ | $73_{\pm2}$ | $.264_{\pm.001}$ | $56_{\pm3}$ |
| Eigen | $73_{\pm3}$ | $40_{\pm2}$ | $79_{\pm2}$ | $75_{\pm3}$ | $89_{\pm3}$ | $69_{\pm12}$ | $39_{\pm6}$ | $72_{\pm4}$ | $74_{\pm2}$ | $.276_{\pm0.002}$ | $61_{\pm2}$ |
| HOSC | $73_{\pm2}$ | $72_{\pm1}$ | $80_{\pm2}$ | $78_{\pm3}$ | $90_{\pm2}$ | $84_{\pm7}$ | $36_{\pm9}$ | $75_{\pm5}$ | $74_{\pm2}$ | $.283_{\pm0.001}$ | $49_{\pm7}$ |
| BN-Pool | $\mathbf{75}_{\pm2}$ | $\mathbf{99}_{\pm0}$ | $\mathbf{81}_{\pm1}$ | $\mathbf{80}_{\pm2}$ | $\mathbf{91}_{\pm2}$ | $\mathbf{88}_{\pm7}$ | $\mathbf{54}_{\pm7}$ | $75_{\pm4}$ | $\mathbf{78}_{\pm1}$ | $\mathbf{.255}_{\pm.001}$ | $58_{\pm2}$ |

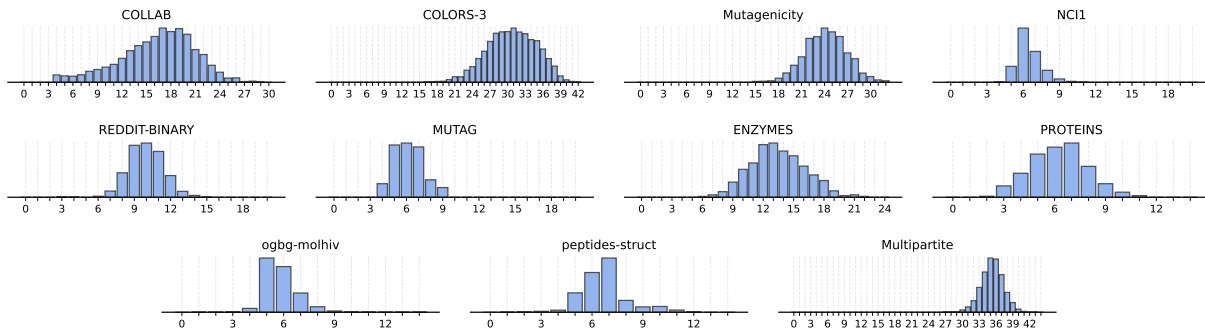

Figure 14: Distribution of non-empty clusters found by BN-Pool on different datasets.

different features in the same clusters (Figure 13c). In particular, MinCut finds 4 clusters even though there are 5 different feature values.

Top-$k$ and $k$-MIS are pooling operators from different families (Score-Based and One-Every-$K$) that pool the graph in different ways. In particular, Top-$k$ (Figure 13d) keeps only half of the nodes and drops the others, shown in black. On the other hand, $k$-MIS does not use the node features, so there is no direct match between the features and the clusters it finds. Figures 13f-i show the different assignment matrices $\boldsymbol{S}$ from these methods, and Figures 13j-m show the topology and node features of the pooled graphs.

BN-Pool uses only 5 clusters that match the 5 types of features. As a result, the pooled graph summarizes effectively the original, with just 5 supernodes, each one tied to a certain feature. On the other hand, MinCut produces a denser assignment matrix $\boldsymbol{S}$, where each node belongs to multiple supernodes, and several supernodes have the same role. This overlap is also visible in the pooled graph, which has many supernodes with similar features. Unlike BN-Pool, this pooled graph is less compact, is very dense, and, thus, more costly to process.

Looking at Top-$k$, we see that its pooled graph is simply a subset of the original, which means some parts of the graph are left out. This is known to be a potential issue in Score-Based methods as it affects their expressivity (Bianchi & Lachi, 2023). Finally, $k$-MIS yields a pooled graph that, like BN-Pool, is both small and very sparse. However, while it represents all parts of the graph, it does not match its supernodes to the node features, as it does not consider them when creating the pooled graph.

Our experimental evaluation mainly focuses on homophilic settings, which is the one BN-Pool and most of the existing pooling methods are designed for. In homophilic graphs such as the one in Figure 13, nodes with the same features are strongly connected with each other, making it perfectly reasonable to assign nodes with

the same features to the same supernode in the pooled graph. To provide a more comprehensive evaluation of our method and its limitations, we also consider a completely heterophilic dataset: the Multipartite dataset. In this dataset, there are ten different types of node features, and each node is connected to *all* the nodes in the graph with a *different* feature. Thus, the topology is adversarial and should be overlooked to solve the task correctly. In such a heterophilic setting, BN-Pool does not perform well: its architecture and the losses implement a homophilic bias that connected nodes should be clustered together.

Table 4: Mean and standard deviation of the non-empty clusters found by BN-Pool on different datasets.

| Dataset | Collab | Colors3 | Mutagenicity | NCI1 | RedditB | MUTAG |
|---|---|---|---|---|---|---|
| | $16.5_{\pm 4.5}$ | $31.0_{\pm 4.1}$ | $24.1_{\pm 2.8}$ | $6.5_{\pm 1.0}$ | $10.0_{\pm 1.5}$ | $5.5_{\pm 1.4}$ |
| Dataset | Enzymes | Proteins | MolHiv | Peptides-struct | Multipartite | |
| | $13.5_{\pm 2.3}$ | $6.3_{\pm 1.5}$ | $5.7_{\pm 1.0}$ | $6.7_{\pm 1.4}$ | $35.4_{\pm 2.1}$ | |

We conclude by noting that all Soft-Clustering methods pool each graph in the same predefined number of supernodes $K$. Instead, BN-Pool does not require specifying $K$ and finds a different $K_i$ for each graph, resulting in a non-trivial distribution of the pooled graphs' sizes. Figure 14 shows the distributions of non-empty clusters found by BN-Pool on different datasets, which gives us further insights about the desired number of pooled nodes in each dataset. The statistics of the non-empty clusters found in each dataset are reported in Table 4.

**Sensitivity Analysis.** In Table 5, we report the validation accuracy, averaged over 10 folds, for different configurations of the hyperparameters $\mu_K$, $\sigma_K$, and $\alpha_{\mathrm{DP}}$.

Table 5: Average validation accuracy for different hyperparameter configurations.

| $\mu_K$ | $\sigma_K$ | $\alpha_{\mathrm{DP}}$ | Enzymes | Colors3 |
|---|---|---|---|---|
| 1 | 0.1 | 0.1 | $59_{\pm 2}$ | $98.6_{\pm 0.4}$ |
| 1 | 0.1 | 1 | $58_{\pm 2}$ | $98.3_{\pm 0.4}$ |
| 1 | 0.1 | 10 | $57_{\pm 3}$ | $98.5_{\pm 0.2}$ |
| 1 | 1 | 0.1 | $59_{\pm 2}$ | $98.8_{\pm 0.2}$ |
| 1 | 1 | 1 | $58_{\pm 2}$ | $98.5_{\pm 0.3}$ |
| 1 | 1 | 10 | $59_{\pm 4}$ | $98.6_{\pm 0.2}$ |
| 10 | 0.1 | 0.1 | $61_{\pm 3}$ | $98.8_{\pm 0.2}$ |
| 10 | 0.1 | 1 | $60_{\pm 3}$ | $98.8_{\pm 0.3}$ |
| 10 | 0.1 | 10 | $60_{\pm 3}$ | $99.1_{\pm 0.2}$ |
| 10 | 1 | 0.1 | $61_{\pm 3}$ | $98.8_{\pm 0.3}$ |
| 10 | 1 | 1 | $60_{\pm 4}$ | $98.8_{\pm 0.3}$ |
| 10 | 1 | 10 | $57_{\pm 4}$ | $99.1_{\pm 0.3}$ |
| 30 | 0.1 | 0.1 | $61_{\pm 3}$ | $99.0_{\pm 0.2}$ |
| 30 | 0.1 | 1 | $61_{\pm 3}$ | $99.1_{\pm 0.3}$ |
| 30 | 0.1 | 10 | $59_{\pm 6}$ | $99.2_{\pm 0.3}$ |
| 30 | 1 | 0.1 | $60_{\pm 1}$ | $99.0_{\pm 0.2}$ |
| 30 | 1 | 1 | $59_{\pm 2}$ | $98.9_{\pm 0.3}$ |
| 30 | 1 | 10 | $58_{\pm 4}$ | $99.3_{\pm 0.2}$ |

As shown in Table 5, the average validation accuracy remains remarkably stable across different configurations. For both Enzymes and Colors3, the accuracy fluctuates only slightly, i.e., the differences are not statistically significant, indicating that the model is robust to changes in these hyperparameters. These results suggest that the proposed method does not require extensive hyperparameter tuning to achieve competitive performance, making it suitable for practical applications where computational resources or tuning time may be limited.

## 5.3 Computational resources

The experiments were performed using seven different servers equipped, respectively, with one Nvidia GeForce RTX 3090 (24GB VRAM), two Nvidia GeForce RTX 4090 (24GB VRAM), two Nvidia RTX A6000 (48GB VRAM), and two Nvidia RTX 6000 Ada Generation (48GB VRAM). In Tables 6 and 7, we report the

Table 6: Maximum usage of the GPU VRAM (Gigabytes) and average training time (seconds per epoch) by different pooling operators on node-level tasks.

| | CiteSeer | | Cora | | DBLP | | PubMed | |
|---|---|---|---|---|---|---|---|---|
| | VRAM (*max GB*) | time (*s/epoch*) | VRAM (*max GB*) | time (*s/epoch*) | VRAM (*max GB*) | time (*s/epoch*) | VRAM (*max GB*) | time (*s/epoch*) |
| DiffPool | 0.59 | 0.91 | 0.49 | 0.78 | 5.43 | 1.22 | 6.52 | 1.16 |
| MinCut | 0.55 | 0.99 | 0.49 | 0.81 | 5.43 | 3.56 | 6.52 | 4.06 |
| DMoN | 0.55 | 0.93 | 0.49 | 0.82 | 5.43 | 1.10 | 6.52 | 1.00 |
| JBGNN | 0.55 | 0.88 | 0.49 | 0.79 | 5.43 | 1.07 | 6.52 | 0.95 |
| BN-Pool | 0.65 | 0.92 | 0.53 | 0.88 | 6.70 | 1.79 | 8.09 | 1.61 |
| SP-BN-Pool | 0.55 | 0.67 | 0.51 | 1.38 | 4.49 | 1.11 | 5.26 | 0.93 |

Table 7: Maximum usage of the GPU VRAM (Gigabytes) and average training time (seconds per epoch) by different pooling operators on graph-level tasks.

| | DD | | ReddetB | | MolHiv | | Pep-struct | |
|---|---|---|---|---|---|---|---|---|
| | VRAM (*max GB*) | time (*s/epoch*) | VRAM (*max GB*) | time (*s/epoch*) | VRAM (*max GB*) | time (*s/epoch*) | VRAM (*max GB*) | time (*s/epoch*) |
| Graclus | 1.05 | 0.48 | 1.06 | 0.47 | 1.00 | 22.74 | 1.03 | 23.88 |
| ECPool | 1.08 | 0.52 | 1.08 | 0.46 | 1.00 | 21.24 | 1.05 | 22.70 |
| $k$-MIS | 1.05 | 0.51 | 1.06 | 0.46 | 1.00 | 23.27 | 1.03 | 21.53 |
| Top-$k$ | 1.08 | 0.48 | 1.07 | 0.46 | 1.00 | 23.61 | 1.03 | 23.78 |
| DiffPool | 2.58 | 0.53 | 6.39 | 0.55 | 1.00 | 22.03 | 1.09 | 22.11 |
| MinCut | 1.84 | 0.49 | 3.78 | 0.60 | 1.00 | 25.52 | 1.07 | 42.51 |
| DMoN | 1.37 | 0.48 | 3.35 | 0.50 | 1.00 | 22.33 | 1.05 | 22.17 |
| JBGNN | 1.45 | 0.46 | 2.43 | 0.47 | 1.00 | 22.83 | 1.05 | 22.72 |
| HOSC | 1.05 | 0.84 | 7.23 | 0.93 | 1.00 | 21.64 | 1.06 | 20.92 |
| BN-Pool | 2.53 | 0.61 | 5.97 | 0.82 | 1.01 | 24.05 | 1.13 | 37.05 |
| SP-BN-Pool | 1.13 | 0.88 | 1.14 | 0.83 | 1.00 | 20.72 | 1.05 | 20.83 |

maximum GPU memory usage and the average time to complete an epoch for a GNN configured with different pooling operators on node- and graph-level tasks, respectively. Pooling methods that pre-compute the coarsening of the graph (e.g., SEP and Eigen) are not considered. Times are measured on an Nvidia GeForce RTX 3090.

Regarding the node-level tasks, on the two largest datasets, DBLP and PubMed, BN-Pool uses approximately 20% more GPU memory. The reason is that BN-Pool uses the binary cross-entropy as the reconstruction loss: during backpropagation, the autograd framework retains the input of the sigmoid to compute the gradient. This extra operation also affects the average training times, but the results are comparable with other methods. Conversely, the sparse variant of BN-Pool (SP-BN-Pool) consistently reduces both memory footprint and training time across all datasets except Cora, being the dataset with the highest edge density. This behavior is expected since the negative edge sampling becomes less effective as the input graph density increases (see Appendix A.4 for details).

In the case of graph-level tasks, we process batches of size 16. We considered the three datasets with the highest average number of vertices (Pep-struct, ReddetB, and DD), the dataset with the highest average number of edges (DD), and the dataset with the highest number of graphs (MolHiv). See Table 10 for the details. Soft-clustering methods use significantly more GPU memory only on DD and ReddetB, as these datasets contain large and sparse graphs. In particular, Diffpool and BN-Pool require additional memory to compute the reconstruction loss over all non-edges. Although the memory footprint remains fully manageable on modern GPUs, SP-BN-Pool drastically reduces memory usage, making it comparable to One-Every-$K$ methods such as Top-$k$. Similarly, the training times of BN-Pool are higher on datasets containing larger

Table 8: Average number (and standard deviation) of epochs used to train the GNN in graph-level tasks when using different poolers.

| | GCB-H | Colors3 | IMDB-BINARY | MolHiv | DD |
|---|---|---|---|---|---|
| Graclus | $1205_{\pm 197}$ | $2266_{\pm 539}$ | $791_{\pm 193}$ | $609_{\pm 55}$ | $557_{\pm 20}$ |
| ECPool | $862_{\pm 196}$ | $1667_{\pm 446}$ | $880_{\pm 242}$ | $607_{\pm 82}$ | $562_{\pm 34}$ |
| $k$-MIS | $703_{\pm 78}$ | $1437_{\pm 449}$ | $620_{\pm 151}$ | $782_{\pm 135}$ | $541_{\pm 13}$ |
| Top-$k$ | $1147_{\pm 686}$ | $1813_{\pm 462}$ | $760_{\pm 323}$ | $845_{\pm 131}$ | $552_{\pm 28}$ |
| DiffPool | $998_{\pm 152}$ | $1777_{\pm 540}$ | $771_{\pm 117}$ | $877_{\pm 262}$ | $1410_{\pm 508}$ |
| MinCut | $1043_{\pm 243}$ | $1389_{\pm 216}$ | $788_{\pm 206}$ | $796_{\pm 193}$ | $379_{\pm 75}$ |
| DMoN | $1941_{\pm 1284}$ | $1140_{\pm 136}$ | $690_{\pm 236}$ | $872_{\pm 259}$ | $581_{\pm 21}$ |
| JBGNN | $2096_{\pm 1405}$ | $879_{\pm 251}$ | $830_{\pm 493}$ | $523_{\pm 4}$ | $607_{\pm 74}$ |
| HOSC | $792_{\pm 90}$ | $1459_{\pm 125}$ | $733_{\pm 306}$ | $1016_{\pm 443}$ | $1139_{\pm 372}$ |
| BN-Pool | $2196_{\pm 523}$ | $2127_{\pm 503}$ | $958_{\pm 133}$ | $800_{\pm 162}$ | $696_{\pm 213}$ |

graphs (e.g., DD and RedditB). Conversely, on MolHiv and Pep-struct, the training times are comparable to those of One-Every-$K$ methods. Again, SP-BN-Pool reduces training time in most cases, except for DD, which is the dataset with the highest edge density. This result highlights that, while reconstructing the full adjacency matrix is negligible for small graphs and enables scalability on datasets with a large number of graphs, the sparse implementation of the reconstruction loss effectively reduces both memory footprint and training time on graphs with low edge density.

In Table 8 we report statistics for the number of epochs needed to train the GNN for the graph-level tasks, configured with the different pooling operators. The statistics are computed over different weights initialization and dataset folds, which creates a degree of variability. As discussed in Section 5.2, we use an early stop with patience 100. We see that the number of epochs required by BN-Pool is comparable to the other methods, with the observed differences being largely attributable to the variability arising from different weight initializations and dataset folds.

## 6 Conclusions

We introduced BN-Pool, an elegant graph pooling method that automatically discovers the number of supernodes in each input graph in a principled way. BN-Pool defines a SBM-like generative process for the adjacency matrix. By specifying a DP prior over the cluster memberships, our model can handle a theoretically unbounded number of clusters, providing flexibility across datasets with graphs of heterogeneous sizes. Due to the probabilistic nature of BN-Pool, training is performed through the variational inference framework. We employ a GNN to approximate the posterior of the node cluster membership, which allows conditioning the posterior on the node and edge features, and the downstream task at hand.

Experiments showed that BN-Pool can effectively find a meaningful number of clusters across unsupervised node clustering, graph classification, and graph regression tasks. This is especially true in the homophilic setting, where the generative model assumptions align naturally with the data. Notably, on two graph classification datasets, it outperforms any other pooling method by a significant margin. We also provided a measurement of the computational resources required by our method. While the $\mathcal{O}(N^2)$ complexity associated with the reconstruction loss is generally not a bottleneck for typical graph-classification datasets, we additionally proposed a sparse implementation that reduces the complexity to $O(E)$, enabling the application of BN-Pool to datasets with substantially larger graphs.

To the extent of our knowledge, this is the first attempt to employ BNP techniques to perform graph pooling. This contribution opens the door to a broader integration of Bayesian methods in graph machine learning and GNNs in particular. The probabilistic framework underlying BN-Pool offers several promising directions for future research. First, it can be extended to heterophilic graphs by modifying the prior on the generative process of the adjacency matrix to capture connectivity patterns that differ from homophily. Second, the approach can be adapted to dynamic graphs, where the input graph evolves. In this setting, the generative

process could be conditioned on previous time steps, following principles similar to hidden Markov models (Beal et al., 2001), thereby enabling temporal dependencies to be modeled in a principled way.

**Acknowledgements**

This work was supported by the Norwegian Research Council project 345017: *RELAY: Relational Deep Learning for Energy Analytics.* We gratefully thank NVIDIA Corporation for the donation of some of the GPUs used in this project. We also wish to thank Davide Bacciu for the initial discussions and for helping to establish the collaboration that made this work possible.

# A    Implementation details

In this section, we show how we implement the key operations in BN-Pool using PyTorch as the backend. The code repository to reproduce this work is openly available[6] and an implementation of BN-Pool is available on Torch Geometric Pool[7] (Abate et al., 2026).

## A.1    Priors and Posteriors Definition

Listing 1 shows how we define the prior and the variational parameters, and how we compute the coarsened graph during the forward pass.

```python
import torch.nn.functional as F
import torch as th
from custom_layers import MLP # A generic MLP

# --- Priors (hyperparameters) ---
# Prior for the Stick Breaking Process
register_buffer('alpha_DP', th.ones(n_clusters - 1) * alpha_DP)

# Prior for the cluster-cluster prob. matrix
register_buffer('sigma_K', th.tensor(sigma_K))
register_buffer('mu_K', mu_K * th.eye(n_clusters, n_clusters) -
                mu_K * (1-th.eye(n_clusters, n_clusters)))

# --- Posteriors (parameters) ---
# Transforms node embeddings into posterior distributions for the sticks (alpha_tilde
    and beta_tilde)
self.MLP = MLP(emb_size, hidden_size, 2*(n_clusters-1), bias=False)

# variational parameters for the connectivity matrix K
self.mu_tilde = th.nn.Parameter(k_init * th.eye(n_clusters, n_clusters) -
                        k_init * (1-th.eye(n_clusters, n_clusters)))

def forward(node_embs, adj):

    N = adj.size(-1)

    # Compute the node assignment matrix S
    S, q_pi = get_S(node_embs)

    # Compute the auxiliary loss
    rec = rec_loss(S, adj)
    pi_kl = pi_prior_loss(q_pi)
    K_kl =  K_prior_loss()
    aux_loss = rec + gamma * pi_kl + K_kl

    # Rescale the loss
    aux_loss = aux_loss / (N * N)

    # Compute the coarsened graph
    x_pool = th.einsum('bnk,bnf->bkf', S, x)
    adj_pool = th.matmul(th.matmul(S.transpose(1, 2), adj), S)

    return aux_loss, adj_pool, x_pool
```

Listing 1: Priors hyperparameters and trainable parameters definition.

In particular, the hyperparameters representing the priors are defined as `buffers` since they are not optimized during the training. Conversely, the variational parameters are defined as `parameters` since they are modified by the training algorithm. The variational parameters $\tilde{\alpha}, \tilde{\beta}$ are not defined explicitly since we compute them by applying an MLP to the node embeddings of size `emb_size` generated by a GNN. The value of `n_clusters` indicates the maximum number of clusters we consider (i.e., the truncation level $C$ of the

---

[6] https://github.com/NGMLGroup/Bayesian-Nonparametric-Graph-Pooling
[7] https://github.com/tgp-team/torch-geometric-pool

posterior approximation), and `k_init` is the value used to initialize the variational parameter $\tilde{\boldsymbol{\mu}}$ (i.e., $\eta_K$ in the main text).

In the `forward` function, we show how BN-Pool computes the coarsened graph and the auxiliary loss. The function has two input parameters: `node_embeddings` represents the node embeddings $\boldsymbol{X}'$ obtained by the previous MP layers, while `adj` represents the adjacency matrix $\boldsymbol{A}$. First, we compute the posterior distributions `q_pi`, and the cluster-assignment matrix S. Then, we use these values to compute the auxiliary loss `aux_loss`, the coarsened graph `adj_pool`, and the pooled node features `x_pool`.

## A.2 Cluster Assignments Computation

Listing 2 shows the key operations in the forward pass of our model: given the node embeddings produced by a GNN, we compute the cluster assignment matrix $\boldsymbol{S}$. The forward pass also computes the variational distributions $q_{\boldsymbol{\pi}}$, which will be useful later to compute the losses.

```python
1  def compute_pi_given_sticks(stick_fractions):
2      # Compute the sticks length given the stick fractions
3      log_v = th.concat([th.log(stick_fractions), th.zeros(*stick_fractions.shape[:-1], 1)
       ], dim=-1)
4      log_one_minus_v = th.concat([th.zeros(*stick_fractions.shape[:-1], 1),
5                                    th.log(1 - stick_fractions)], dim=-1)
6      pi = th.exp(log_v + th.cumsum(log_one_minus_v, dim=-1))
7      return pi # has shape: [T, batch, N, C]
8
9  def get_S(node_embs, n_particles, n_clusters):
10     # Compute soft cluster assignments.
11     out = th.clamp(F.softplus(self.MLP(node_embs)), min=1e-3, max=1e3)
12     alpha_tilde, beta_tilde = th.split(out, n_clusters-1, dim=-1)
13     q_pi = th.distributions.Beta(alpha_tilde, beta_tilde)
14     stick_fractions = q_pi.rsample([n_particles])
15     S = compute_pi_given_sticks(stick_fractions)
16     return S, q_pi
```

Listing 2: Forward computation of the cluster assignments.

First, we obtain the variational parameters $\tilde{\boldsymbol{\alpha}}, \tilde{\boldsymbol{\beta}}$ by applying the MLP to the node embeddings produced by the GNN. Note that both variational parameters should be greater than 0; thus, we apply the `softplus` activation function. Moreover, to avoid numerical errors, we clamp the values between $10^{-3}$ and $10^3$.

Once we have the variational parameters, we define the variational distribution by employing the PyTorch class `torch.distributions.Beta`. Then, we sample `n_particles` (i.e., $T$ in the main text) values that will be used to approximate the reconstruction loss by using the `rsample` method. The `r` in the `rsample` name stands for *reparameterization*, that is, the trick that separates the distribution parameters from the randomness and allows to backpropagate the gradient from the samples to the distribution parameters. This technique is also denoted as a *pathwise gradient estimator*. As we mentioned in Section 3.4, the reparameterization trick cannot be applied to the Beta distribution explicitly. Therefore, we rely on an approximation of the pathwise derivative (Figurnov et al., 2018; Jankowiak & Obermeyer, 2018), which does not require reparameterizing the Beta distribution explicitly. This approximation is already implemented in the PyTorch framework: when we call the `rsample` method, the backend computes the pathwise derivative (if possible) or approximates it (as in our case). Thus, the gradient flows from the reconstruction loss to the variational parameters $\tilde{\boldsymbol{\alpha}}, \tilde{\boldsymbol{\beta}}$, and then to the GNN parameters $\Theta$.

The function `compute_pi_given_sticks` computes the stick length $\pi_1, \ldots, \pi_C$ given the stick fractions $\pi'_1, \ldots, \pi'_C$ by applying Equation 4. The computation is performed in the log-space to avoid numerical errors.

## A.3 Losses Computation

Listing 3 shows the computation of the losses $\mathcal{L}_{\mathrm{rec}}, \mathcal{L}_{\boldsymbol{\pi}}, \mathcal{L}_{\boldsymbol{K}}$. The function `rec_loss` computes the reconstruction loss $\mathcal{L}_{\mathrm{rec}}$. As shown in Equation 13, the value of the loss corresponds to the Binary Cross-Entropy (BCE) loss computed between the adjacency matrix $\boldsymbol{A}$ and the probability to have an edge for each node pair. Note

that we use `BCE_with_logits` rather than applying the sigmoid function to each $\boldsymbol{\pi}_u^\top \tilde{\boldsymbol{\mu}} \boldsymbol{\pi}_v$. Since the number of edges is usually much smaller than the total number of possible edges, we assign different weights to the positive and negative classes to achieve class balancing. The weights for the positive class are computed in line 10 and stored in the variable `balance_weights`.

The loss $\mathcal{L}_{\boldsymbol{\pi}}$ is equal to the KL divergence between the prior $p(\boldsymbol{\pi}'_{ui})$ and the variational posterior $q(\boldsymbol{\pi}'_{ui})$ for each node $u$ and a cluster $i$. Since all the distributions involved are Beta distributions, the KL divergence has a closed form, and it is already implemented in PyTorch. This loss is computed by the function `pi_prior_loss`.

```python
def rec_loss(S, A):

    # Compute the percentage of non-zero links
    # N is the number of nodes
    # E is the number of edges
    balance_weights = (N*N / E) * A + (N*N / (N*N - E)) * (1 - A)

    # Compute the probability of an edge for each node pair, i.e. S K S^T
    p_adj = S @ self.mu_tilde @ S.transpose(-1,-2)

    loss = F.binary_cross_entropy_with_logits(p_adj, A, weight=balance_weights,
    reduction='none')

    return loss

def pi_prior_loss(self, q_pi):
    alpha_DP = self.get_buffer('alpha_DP')
    p_pi = Beta(th.ones_like(alpha_DP), alpha_DP)
    loss = kl_divergence(q_pi, p_pi).sum(-1)
    return loss

def K_prior_loss(self):
    mu_K, sigma_K = self.get_buffer('mu_K'), self.get_buffer('sigma_K')
    K_prior_loss = (0.5 * (self.mu_tilde - mu_K) ** 2 / sigma_K).sum()
    return K_prior_loss
```

Listing 3: Losses computation.

The last loss $\mathcal{L}_{\boldsymbol{K}}$ is equal to the KL divergence between normal distributions since $q(\boldsymbol{K}_{ij})$ and $p(\boldsymbol{K}_{ij})$ are Gaussian distributions for all clusters $i$ and $j$. Since we do not optimize the variance of the variational distribution, we can ignore all the terms that do not involve the variational parameters $\tilde{\boldsymbol{\mu}}$. Thus, we compute $\mathcal{L}_{\boldsymbol{K}}$ as the mean squared error between $\tilde{\boldsymbol{\mu}}$ and $\mu_{\boldsymbol{K}}$ scaled by the variance prior $\sigma_{\boldsymbol{K}}$. This loss is computed by the function `K_prior_loss`.

## A.4 Sparse Computation of the Reconstruction Loss

The implementation of the sparse reconstruction surrogate defined in equation 14 relies on efficient negative edge sampling. The goal is to generate non-existent edges (i.e., the "negative edges") that are used in the loss computation, avoiding the materialization of the full adjacency matrix. The efficiency of negative edge sampling relies on balancing memory usage and computational cost, depending on the sparsity of the graph. The critical factor is the relationship between the number of observed edges ($E$) and the total number of possible edges ($N^2$).

For *sparse graphs* ($E \ll N^2$), where the number of edges is much smaller than the total possible edges, it is highly likely that a randomly sampled edge is a negative one, i.e., the edge is not present in the original graph. Thus, we can efficiently sample negative edges by randomly generating node pairs without explicitly enumerating all possible edges. This approach has a significantly lower memory complexity and computational cost, as it operates only on the existing edges and avoids creating a dense adjacency structure.

As the graph density increases and $E$ approaches $N^2$, the probability of randomly sampling a negative edge decreases. In this case, producing $E$ negative samples may require $\mathcal{O}(N^2)$ sampling trials. Moreover, for dense graphs, the memory complexity of storing $E$ edges is already $\mathcal{O}(N^2)$, meaning that the additional memory cost in explicitly enumerating all possible edges is lower. Therefore, in such scenarios, we explicitly

enumerate all possible edges, excluding the observed ones, and directly sample from the remaining set of negative edges.

To determine whether to use sparse sampling or dense enumeration, we employ a heuristic considering the sparsity of the input graph. The heuristic operates on the probability of sampling a valid negative edge, estimated as $1 - (E/N^2)$. If the probability is above a threshold (e.g., 50%), sparse sampling is used; otherwise, the dense approach is preferred. This adaptive mechanism ensures an optimal trade-off between computational and memory requirements.

All the operations are performed directly on the GPU to avoid unnecessary data transfer.

## B   Datasets details

The details of the datasets used in the node clustering task are reported in Table 9. We also report the intra-class and inter-class density, which is the average number of edges between nodes that belong to the same class or to different classes, respectively. The Community dataset is generated using the PyGSP library[8]. The other datasets are obtained with the PyG loaders[9].

Table 9: Details of the vertex clustering datasets.

| Dataset | Vertices | Edges | Vertex attr. | Vertex classes | Intra-class density | Inter-class density |
|---|---|---|---|---|---|---|
| Community | 400 | 5,904 | 2 | 5 | 0.1737 | 0.0025 |
| Cora | 2,708 | 10,556 | 1,433 | 7 | 0.0065 | 0.0004 |
| CiteSeer | 3,327 | 9,104 | 3,703 | 6 | 0.0034 | 0.0003 |
| PubMed | 19,717 | 88,648 | 500 | 3 | 0.0005 | 0.0001 |
| DBLP | 17,716 | 105,734 | 1,639 | 4 | 0.0008 | 0.0001 |

Table 10: Details of the graph classification datasets.

| Dataset | Samples | Classes | Avg. vertices | Avg. edges | Vertex attr. | Vertex labels | Edge attr. |
|---|---|---|---|---|---|---|---|
| GCB-H | 1,800 | 3 | 148.32 | 572.32 | – | yes | – |
| Collab | 5,000 | 3 | 74.49 | 4,914.43 | – | no | – |
| Colors3 | 10,500 | 11 | 61.31 | 91.03 | 4 | no | – |
| IMDB | 1,000 | 2 | 19.77 | 96.53 | – | no | – |
| Mutagenicity | 4,337 | 2 | 30.32 | 61.54 | – | yes | – |
| NCI1 | 4,110 | 2 | 29.87 | 64.60 | – | yes | – |
| RedditB | 2000 | 2 | 429.63 | 497.75 | – | no | – |
| D&D | 1,178 | 2 | 284.32 | 1,431.32 | – | yes | – |
| MUTAG | 188 | 2 | 17.93 | 19.79 | – | yes | – |
| Proteins | 1,113 | 2 | 39.06 | 72.82 | 1 | yes | – |
| Enzymes | 600 | 6 | 32.63 | 62.14 | 18 | yes | – |
| MolHiv | 41,127 | 2 | 25.5 | 27.5 | 9 | no | 3 |
| Pep-struct | 15,535 | – | 150.9 | 307.3 | 9 | no | 3 |
| Multipartite | 5,000 | 10 | 99.79 | 4,477.43 | – | yes | 3 |

The details of the datasets used in the graph classification task are reported in Table 10. All datasets besides GCB-H, Multipartite, Pep-struct, and ogbg-molhiv are downloaded from the TUDataset repository[10] using the PyG loader. For the GCB-H, we used the data loader provided in the original repository[11]. MolHiv is obtained from the OGB repository[12] through the loader from the OGB library. The Multipartite dataset is

---

[8]https://pygsp.readthedocs.io/en/latest/

[9]https://pytorch-geometric.readthedocs.io/en/latest/modules/datasets.html

[10]https://chrsmrrs.github.io/datasets/

[11]https://github.com/FilippoMB/Benchmark_dataset_for_graph_classification

[12]https://ogb.stanford.edu/docs/graphprop/

obtained from the original repository[13], and Pep-struct is downloaded using the PyG loader for the Long Range Graph Benchmark datasets[14].

## B.1   Hyperparameters configuration

In Table 11 we report the optimal configuration of $\alpha_{\mathrm{DP}}$, $\mu_K$, and $\sigma_K$ in each dataset, i.e., those that yield the best classification accuracy, MSE, or AUROC on the validation set.

Table 11: Optimal configuration of the hyperparameters of BN-Pool for each dataset.

| Dataset | $\alpha_{\mathrm{DP}}$ | $\mu_K$ | $\sigma_K$ |
|---|---|---|---|
| GCB-H | 10.0 | 30.0 | 1.0 |
| Collab | 1.0 | 10.0 | 0.1 |
| Colors3 | 10.0 | 30.0 | 1.0 |
| IMDB | 10.0 | 10.0 | 0.1 |
| Mutagenicity | 10.0 | 30.0 | 1.0 |
| NCI1 | 10.0 | 1.0 | 0.1 |
| RedditB | 1.0 | 1.0 | 1.0 |
| D&D | 30.0 | 1.0 | 1.0 |
| MUTAG | 1.0 | 10. | 1.0 |
| Proteins | 1.0 | 1.0 | 0.1 |
| Enzymes | 1.0 | 30.0 | 0.1 |
| MolHiv | 10.0 | 1.0 | 0.1 |
| Pep-struct | 1.0 | 0.1 | 0.1 |
| Multipart. | 1.0 | 10.0 | 1.0 |

## C   Additional experiments

In Table 12 we report the additional results that were omitted from the main body.

Table 12: Mean and standard deviations of the graph classification accuracy.

| Pooler | GCB-H | IMDB | DD |
|---|---|---|---|
| Graclus | $\mathbf{75}_{\pm 3}$ | $\mathbf{77}_{\pm 6}$ | $73_{\pm 4}$ |
| ECPool | $\mathbf{75}_{\pm 4}$ | $75_{\pm 7}$ | $73_{\pm 5}$ |
| $k$-MIS | $\mathbf{75}_{\pm 4}$ | $74_{\pm 7}$ | $75_{\pm 3}$ |
| Top-$k$ | $56_{\pm 5}$ | $74_{\pm 5}$ | $72_{\pm 5}$ |
| SEP | $74_{\pm 2}$ | $72_{\pm 6}$ | $77_{\pm 3}$ |
| DiffPool | $51_{\pm 8}$ | $72_{\pm 6}$ | $75_{\pm 4}$ |
| MinCut | $\mathbf{75}_{\pm 5}$ | $73_{\pm 6}$ | $78_{\pm 5}$ |
| DMoN | $74_{\pm 3}$ | $73_{\pm 6}$ | $78_{\pm 5}$ |
| JBGNN | $\mathbf{75}_{\pm 4}$ | $75_{\pm 6}$ | $79_{\pm 4}$ |
| Eigen | $62_{\pm 2}$ | $71_{\pm 6}$ | $74_{\pm 4}$ |
| HOSC | $\mathbf{75}_{\pm 2}$ | $74_{\pm 5}$ | $79_{\pm 2}$ |
| BN-Pool | $\mathbf{75}_{\pm 3}$ | $76_{\pm 5}$ | $\mathbf{80}_{\pm 3}$ |

In Table 13 we report the results obtained by other soft-clustering methods by sweeping the number of supernodes $K$ over the values $0.25\bar{N}$ and $0.1\bar{N}$.

---

[13]https://zenodo.org/records/11617423
[14]https://pytorch-geometric.readthedocs.io/en/latest/generated/torch_geometric.datasets.LRGBDataset.html

Table 13: Mean and standard deviations of the graph classification accuracy (ROC-AUC for MolHiv and MAE for Pep-struct) when using different pooling ratios $K = 0.25\bar{N}$ and $K = 0.1\bar{N}$.

| Pooler | Collab | Colors3 | Mutagenicity | NCI1 | RedditB | MUTAG | Enzymes | Proteins | MolHiv | Pep-struct | Multip. |
|---|---|---|---|---|---|---|---|---|---|---|---|
| DiffPool (0.25) | $65_{\pm 3}$ | $56_{\pm 3}$ | $79_{\pm 2}$ | $73_{\pm 3}$ | $78_{\pm 2}$ | $87_{\pm 8}$ | $21_{\pm 6}$ | $73_{\pm 5}$ | $73_{\pm 2}$ | $.330_{\pm 0.010}$ | $54_{\pm 2}$ |
| DiffPool (0.1) | $66_{\pm 5}$ | $65_{\pm 4}$ | $80_{\pm 2}$ | $71_{\pm 5}$ | $77_{\pm 7}$ | $79_{\pm 10}$ | $20_{\pm 4}$ | $71_{\pm 4}$ | $68_{\pm 2}$ | $.341_{\pm 0.019}$ | $58_{\pm 1}$ |
| MinCut (0.25) | $71_{\pm 2}$ | $68_{\pm 2}$ | $81_{\pm 2}$ | $78_{\pm 3}$ | $90_{\pm 1}$ | $86_{\pm 8}$ | $34_{\pm 5}$ | $74_{\pm 6}$ | $75_{\pm 3}$ | $.270_{\pm 0.003}$ | $58_{\pm 5}$ |
| MinCut (0.1) | $70_{\pm 2}$ | $75_{\pm 1}$ | $80_{\pm 2}$ | $78_{\pm 2}$ | $90_{\pm 2}$ | $85_{\pm 9}$ | $36_{\pm 6}$ | $74_{\pm 6}$ | $72_{\pm 5}$ | $.288_{\pm 0.007}$ | $60_{\pm 3}$ |
| DMoN (0.25) | $69_{\pm 3}$ | $72_{\pm 1}$ | $82_{\pm 3}$ | $79_{\pm 2}$ | $90_{\pm 1}$ | $88_{\pm 8}$ | $35_{\pm 8}$ | $73_{\pm 5}$ | $76_{\pm 2}$ | $.277_{\pm 0.003}$ | $55_{\pm 5}$ |
| DMoN (0.1) | $70_{\pm 1}$ | $69_{\pm 1}$ | $80_{\pm 1}$ | $78_{\pm 3}$ | $91_{\pm 2}$ | $85_{\pm 9}$ | $40_{\pm 9}$ | $74_{\pm 3}$ | $75_{\pm 0}$ | $.306_{\pm 0.014}$ | $62_{\pm 2}$ |
| JBGNN (0.25) | $71_{\pm 1}$ | $69_{\pm 1}$ | $80_{\pm 2}$ | $79_{\pm 3}$ | $92_{\pm 1}$ | $88_{\pm 6}$ | $40_{\pm 5}$ | $74_{\pm 5}$ | $75_{\pm 0}$ | $.317_{\pm 0.003}$ | $51_{\pm 3}$ |
| JBGNN (0.1) | $70_{\pm 2}$ | $67_{\pm 2}$ | $80_{\pm 2}$ | $77_{\pm 3}$ | $92_{\pm 1}$ | $85_{\pm 9}$ | $37_{\pm 7}$ | $73_{\pm 4}$ | $76_{\pm 1}$ | $.317_{\pm 0.004}$ | $59_{\pm 3}$ |
| HOSC (0.25) | $72_{\pm 2}$ | $77_{\pm 1}$ | $80_{\pm 2}$ | $77_{\pm 2}$ | $90_{\pm 1}$ | $87_{\pm 5}$ | $37_{\pm 7}$ | $74_{\pm 5}$ | $76_{\pm 1}$ | $.277_{\pm 0.004}$ | $24_{\pm 3}$ |
| HOSC (0.1) | $70_{\pm 1}$ | $78_{\pm 1}$ | $80_{\pm 2}$ | $77_{\pm 2}$ | $91_{\pm 1}$ | $87_{\pm 5}$ | $36_{\pm 8}$ | $74_{\pm 6}$ | $75_{\pm 2}$ | $.285_{\pm 0.006}$ | $18_{\pm 8}$ |

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
