# OpenReview forum: "BN-Pool: Bayesian Nonparametric Pooling for Graphs"
_TMLR — Accepted by TMLR_

### Review · Reviewer_h9KQ · 2025-12-16

**Summary Of Contributions:**

This work propose BN-Pool, the first soft-clustering pooling operator capable of adaptively determining the number of supernodes for each individual graph. Unlike previous methods that rigidly fix # of supernodes or set it as a fraction of graph size, BN-Pool utilizes a Bayesian Non-parametric framework with a Dirichlet Process prior to allow for a theoretically unbounded number of clusters. This is implemented via a variational autoencoder where a GNN approximates the posterior of node-to-cluster assignments, optimized by combining downstream task loss with an auxiliary reconstruction loss.

In node clustering tasks, e.g., community detection, this work effectively identifies communities and interaction patterns without prior knowledge of the number of classes, reconstructing adjacency matrices that better reflect the original graph sparsity compared to methods like MinCut.

In graph classification, the method performs on par with or superior to existing pooling operators, achieving state-of-the-art results on datasets such as Colors-3 and Enzymes. BN-Pool avoids the redundancy typical of other soft-clustering methods by generating compact pooled graphs that preserve informative content while discarding unnecessary clusters.

**Audience:**

Yes

**Audience Explanation:**

This article is based on the classic DP scheme for graph clustering. This simple combination of traditional schemes with neural networks should be of interest to the audience.

**Broader Impact Concerns:**

there's no broader impact concerns.

**Claims And Evidence:**

Yes

**Claims Explanation:**

Theoretically,  the core evidence for the model's ability to adaptively determine the # of nodes comes from the properties of DP, which dictates that even with infinite capacity, the process will favor reusing existing components, and using the Stick-Breaking Process to construct the mixing proportions.

Additionally, this work demonstrates effectiveness from multiple experimental perspectives, e.g., adaptive clutering ability, competitive performance and compact representations.

**Requested Changes:**

The experimental validation would benefit from a more rigorous analysis of the model's robustness and the specific contributions of its components. Specifically, the paper lacks a sensitivity analysis for the hyperparameter $\eta$ in Equation 12, which balances the reconstruction loss and clustering penalty; it is crucial to demonstrate whether extreme values of $\eta$ cause the model to degenerate into trivial solutions, such as assigning all nodes to a single cluster. Furthermore, since BN-Pool introduces three distinct loss terms, an explicit ablation study is necessary to quantify the independent performance contribution of each term. Finally, while the paper highlights successful clustering on datasets like GCB-H, the discussion should be expanded to include failure cases or specific scenarios where the model struggles, such as on heterophilic graphs, to provide a more comprehensive evaluation of its limitations.

Regarding complexity and scalability, the reliance on an $O(N^2)$ reconstruction loss poses a clear challenge for scaling to larger graphs, despite the authors' assertion that it is manageable for current benchmarks. The paper should discuss or propose specific optimization strategies to mitigate this, such as employing edge sampling or negative sampling to approximate the loss and potentially reduce complexity to $O(|E|)$ or $O(N)$. Additionally, the current runtime comparison relies solely on time-per-epoch, which may be misleading; given the complex variational inference and auxiliary losses involved, BN-Pool likely requires more epochs to converge than simpler baselines. Therefore, comparing the total time required to reach convergence would offer a more realistic and transparent assessment of the training costs.

---

> ### Author Response · Authors · 2026-02-11
>
> ## Sensitivity of $\eta$
>
> We thank the reviewer for highlighting the importance of the hyperparameter balancing the reconstruction loss and the clustering penalty.
>
> First, we would like to clarify that in the revised manuscript, **we have renamed this hyperparameter from $\eta$ to $\gamma$ (Equation 12)** to avoid any confusion with the initialization parameter $\eta_\mathcal{K}$.
>
> Regarding the sensitivity analysis, we wish to clarify that $\gamma$ is not a fixed hyperparameter that we tune, but rather a value that we dynamically schedule during training. As detailed in Section 5.1 of the revised paper, we increase $\gamma$ from 0 to 1 over the first 5,000 epochs using a cosine scheduler and from 0 to 1 over 50 epochs for the graph-level tasks (Section 5.2).
>
> This procedure, widely known in the literature as KL annealing [1, 2], is a standard practice when training Variational Autoencoders. Its specific purpose is to prevent the model from converging to trivial solutions (such as posterior collapse or assigning all nodes to a single cluster) early in the training process, which is exactly the concern raised by the reviewer. By starting with $\gamma=0$, the model initially focuses on learning to reconstruct the graph structure; as $\gamma$ effectively increases, the regularization terms are gradually introduced, forcing the latent structure to conform to the priors without destroying the learned representations.
>
> Since $\gamma$ varies across its entire meaningful range [0, 1] during every training run, a sensitivity analysis on a static value would not reflect the actual training dynamics of our method.
>
> References:
>
> [1] Bowman, S. R., et al. "Generating sentences from a continuous space." CoNLL 2016.
>
> [2] Sønderby, C. K., et al. "Ladder variational autoencoders." NeurIPS 2016.
>
>
> ## Ablation on the terms of the auxiliary losses
>
> We would like to respectfully clarify that the three loss terms ($\mathcal{L}\_{\text{rec}}$, $\mathcal{L}\_{\pi}$, $\mathcal{L}\_{\mathcal{K}}$) in our objective function are **not** independent heuristics that can be arbitrarily removed or combined. Rather, they are mathematically derived components that collectively constitute the ELBO of our variational inference framework.
>
> Specifically:
>
> - $\mathcal{L}_{\text{rec}}$ is the reconstruction term.
> - $\mathcal{L}\_{\pi}$ and $\mathcal{L}\_{\mathcal{K}}$ are the KL-divergence terms for the latent variables.
>
> Removing any single term would mathematically invalidate the derivation of the variational lower bound and break the probabilistic foundation of the model. For instance, removing the KL term on $\pi$ would mean that the posterior is no longer constrained by the DP prior, vanishing the effect of the hyper-parameter $\alpha_{\text{DP}}$ to control the number of clusters used.
>
>
> We have now explicitly stated this in Section 5.1 of the revised manuscript:
>
> > "It is important to note that these terms are not independent auxiliary objectives but are intrinsically linked as they constitute the ELBO. Therefore, removing one of these terms would not be meaningful, as it would invalidate the variational lower bound derivation and the probabilistic foundation of the model."
>
> We hope this clarifies why a standard ablation study is not meaningful in this specific context.
>
>
> ## Inclusion of failure cases
>
> Thanks for the useful suggestion.
>
> We included a new experiment on the Multipartite dataset from [3]. The dataset is completely heterophilic: there are ten different types of node features, and each node is connected to \textit{all} the nodes in the graph with a \textit{different} feature. In this dataset, the topology is adversarial and should be overlooked to solve the task correctly. In the revised manuscript, we show that on this dataset BNPool struggles, since its architecture and the losses implement a homophilic bias that connected nodes should be clustered together.
>
> Nevertheless, the bias toward homophily is explicitly encoded in the generative model. By modifying this model (e.g., by adjusting the prior on inter-cluster connectivity) we can adapt BN-Pool to heterophilic settings without changing the overall framework. This flexibility is a key advantage of our probabilistic formulation of the pooling operator.
>
> References:
>
> [3] Abate & Bianchi, MaxCutPool: differentiable feature-aware Maxcut for pooling in graph neural networks, ICLR 2025.

---

> ### Author Response · Authors · 2026-02-11
>
> ## Mitigation of algorithmic complexity with edge sampling
>
> We agree with the reviewer that, although the current $O(N^2)$ complexity is consistent with other dense pooling methods, there is room for improvement by adopting edge negative sampling.
>
> To make the algorithm complexity proportional to the number of edges, **we implemented a sparse variant** where a subset of the negative examples (i.e., the non-edges) is randomly sampled. The key idea is to replace the summation over all possible node pairs $(u,v)$ in the reconstruction loss (Eq. 13). Instead of computing the binary cross-entropy for every pair of nodes, we compute it only for the actual edges in the graph (positive examples, which are $E$ in total) and for a subset of non-edges $(u,v)$ (negative examples), so that $(u,v) \notin \mathcal{E}$. This reduces the overall complexity to $O(E)$, which is typically much smaller than $O(N^2)$ for large graphs (see Table 10).
>
> We notice that the proposed variant trades the lower memory complexity for potentially higher compute time and is particularly effective only when the graphs are sparse and very large. Indeed, in some settings where these properties are not reflected in the data, the dense version is more efficient.
> While this modification is theoretically straightforward, in practice, we encountered challenges in efficiently performing negative edge sampling on GPUs. In the revised manuscript, we discuss both the theoretical aspects and practical implementation of this sparse version of BN-Pool, along with its impact on memory and training time for datasets such as DD, Reddit-B, Molhiv, and Pep-struct.
>
> ## More realistic and transparent assessment of the training costs
>
> Thanks for raising this important point.
>
> We added to the revised manuscript a new table showing the average number (and std.) of epochs needed to train the GNN configured with the different pooling operators. The new results and discussion are reported in section 5.3 of the revised manuscript.

---

### Review · Reviewer_21r7 · 2025-12-18

**Summary Of Contributions:**

The paper introduces BN-Pool, a novel graph pooling operator for Graph Neural Networks (GNNs) based on Bayesian nonparametric modeling. Its core contribution is addressing a key limitation of existing soft-clustering pooling methods (e.g., DiffPool, MinCut): the need to predefine a fixed number of clusters (supernodes) for all graphs, thereby achieving superior performance.

(+) Adaptive and data-driven pooling that eliminates the need to tune or guess the number of clusters or pooling ratios

(+) Strong theoretical grounding in the Bayesian nonparametric framework

(-) Additional probabilistic hyperparameters ($\alpha_{DP}$, $\mu_K$, $\sigma_K$) make it unclear whether there is actually a benefit

(-) The presentation is unclear, making it difficult to understand the details of the proposed method

**Audience:**

Yes

**Audience Explanation:**

GNN and pooling layers are classic topics in machine learning and will be of interest to at least some individuals in TMLR's audience.

**Claims And Evidence:**

No

**Claims Explanation:**

My major concerns are two-fold:
- While the proposed method removes the need to explicitly tune the number of clusters, it is unclear whether this constitutes a net practical benefit. In particular, BN-Pool introduces several additional hyperparameters (e.g., $\alpha_{DP}$, $\mu_K$, and $\sigma_K$), whose selection may be non-trivial and could potentially offset the advantage of eliminating the cluster-count hyperparameter. As a result, the overall complexity of model tuning may not be reduced and could even increase.
This issue also raises concerns regarding the experimental comparison: the proposed hyperparameters are tuned for BN-Pool, whereas the number of clusters for competing pooling methods appears to be fixed rather than similarly optimized. This asymmetry makes it difficult to attribute the reported performance gains to the proposed methodology itself, rather than to differences in hyperparameter tuning effort.
- The presentation is also unclear. There is no clear definition of the proposed BN-Pool layer in Section 3: what are the inputs and outputs of the layer? What are its hyperparameters? How do one apply it to an existing GNN? How about the additional GNN? I also find there is a gap between the BNP theory and the proposed method and do not quite understand how it is used to design BN-Pool in detail.

**Requested Changes:**

As mentioned above, I request changes in: 1) clarifying the benefits of the proposed method in terms of the additional hyperparameters introduced; 2) revising the experiments to ensure fairness against hyperparameter tuning and their burden, together with analysis/explanations of the new results; 3) rewriting the methodology part to facilitate understanding of the proposed method.

---

> ### Author Response · Authors · 2026-02-11
>
> ## Sensitivity of the hyperparameters
>
> We thank the reviewer for raising this important point regarding the practical trade-off between tuning hyperparameters and fixing the number of clusters.
>
> While it is true that BN-Pool introduces new hyperparameters ($\alpha_\text{DP}, \mu_\mathcal{K}, \sigma_\mathcal{K}$), we respectfully disagree that this increases the overall complexity or limits the benefits of our approach. We base this on three main arguments:
>
> - As demonstrated in our Sensitivity Analysis (Table 5), the performance of BN-Pool is remarkably stable across different configurations of these hyperparameters. The fluctuations in accuracy are minor and not statistically significant. This indicates that these hyperparameters are not "critical" values that require fine-grained tuning (unlike the number of clusters $K$ in other methods, where a wrong choice can lead to collapse or loss of information). Indeed, our hyperparameters define *distributions* (soft priors) rather than hard constraints, making the model inherently more flexible.
> - The core innovation of BN-Pool is not just finding the "best" average number of clusters for a dataset, but adaptively determining the number of clusters $K_i$ for each individual graph $i$. Even if one were to perfectly tune the fixed $K$ for a competing method like MinCut or DiffPool, those methods would still be constrained to pool every graph to the same size (or ratio). This is a structural limitation that tuning cannot solve, and it is exactly what BN-Pool addresses by allowing the complexity of the pooled graph to vary with the input.
> - Regarding the experimental comparison, we followed the standard protocol in the literature for competing methods by obtaining the number of clusters $K$ from the data itself (setting $K$ as a fraction of the average number of nodes, $K = 0.5\bar{N}$). This is the standard "rule of thumb" used when $K$ is not treated as a hyperparameter to be grid-searched and **is done in the original papers** proposing the competing methods. In contrast, while we did perform model selection for BN-Pool, the sensitivity analysis suggests that a default configuration would likely have yielded comparable results, mitigating concerns about unfair advantages due to tuning effort.  Finally, we conducted additional experiments to ensure a fair comparison, sweeping over different values of K for soft‑clustering baselines. The results are in Appendix C.
>
> Finally, we note that the Bayesian framework allows for even further abstraction: if one wished to avoid setting specific values for $\mu_\mathcal{K}$ or $\alpha_\text{DP}$, it is theoretically possible to place hyper-priors on them, letting the data drive their selection entirely, though we found this unnecessary given the method's robustness.
>
>
> ## Clarity of the presentation
>
> We thank the reviewer for the suggestions to improve the clarity of the presentation. We made several modifications:
>
> - We significantly improved the description of the BN-Pool layer in the **methodology** section 3 by adding additional explanation, restructuring the organization of the subsections, and by improving Figure 7 (and its caption) to be more detailed and precise. We also better clarified the architecture of the GNN used to generate the variational parameters $\tilde{\alpha}$ and $\tilde{\beta}$ (see Eq. 10 and accompanying text).
> - In the **background**, we added Figure 4, which shows the difference between a flat and a hierarchical GNN architecture. We also added Figure 5, which illustrates a schematic depiction of a graph pooling layer (with an auxiliary loss) and shows how the SEL, RED, and CON operations interact with each other.
> - In **appendix A**, we expanded Listing 1 to show the additional operations in the forward pass of the BN-Pool layer.

---

### Review · Reviewer_V28m · 2026-02-04

**Summary Of Contributions:**

This paper proposes BN-Pool, a graph pooling operator that uses a Bayesian nonparametric formulation to adaptively choose the number of supernodes per input graph, rather than fixing a pooled size K across graphs. The method defines an SBM-like generative process for edges based on latent soft cluster memberships and a cluster-cluster connectivity matrix. Training combines the downstream supervised loss with an auxiliary variational objective derived from the ELBO. Empirically, the paper evaluates (i) unsupervised node clustering and (ii) graph classification/regression, reporting competitive or improved results compared to existing pooling operators on multiple benchmarks.

Strengths:
1. Clear, principled formulation that ties pooling to a probabilistic generative model and explains how adaptive K emerges.
2. Broad experimental scope across node clustering and graph-level prediction, plus sensitivity and resource reporting.

Weaknesses:
1. Experiments can be improved; the hyperparameters of baselines can be tuned better, and more competitive pooling methods can be added.
2. Claims about compactness/redundancy are not directly quantified in the main evaluations; the evidence focuses primarily on predictive performance.

**Audience:**

Yes

**Audience Explanation:**

Graph pooling is a central component in hierarchical GNNs, and choosing K is a persistent practical and conceptual pain point. BN-Pool provides a principled way to make pooled size data-dependent per graph (and potentially task-dependent through joint training), which is likely of interest to readers working on graph representation learning, probabilistic deep learning, and scalable/robust GNN architectures. The empirical results suggest BN-Pool can match or exceed existing pooling methods on standard benchmarks.

**Broader Impact Concerns:**

No ethical issues are found.

**Claims And Evidence:**

Yes

**Claims Explanation:**

The method definition directly encodes the adaptive number of supernodes via a DP prior and a variational objective in which the KL term on stick-breaking variables functions as a cluster-usage regularizer; this is clearly described and mathematically grounded. Also the result tables support the competitiveness of the proposed method on various graph-level tasks. The paper includes GPU memory and time-per-epoch comparisons and concludes that the O(N^2) complexity is generally not limiting on typical benchmarks.

**Requested Changes:**

1. Add quantitative measures tied to the “compactness” claim: e.g., distribution of learned K_i per dataset, average pooled graph density/sparsity, etc.
2. For more comprehensive comparison, consider adding (a) tuned K sweeps for soft-clustering baselines and (b) tuned \kappa sweeps for score-based baselines
3. Graph pooling was a popular area of study in graph neural networks. Compare with more pooling methods:
[1] https://arxiv.org/abs/1904.13107
[2] https://proceedings.mlr.press/v162/wu22b.html
[3] https://dl.acm.org/doi/abs/10.1145/3511808.3557353
[4] https://ojs.aaai.org/index.php/AAAI/article/view/25866

---

> ### Author Response · Authors · 2026-02-11
>
> ## Quantitative Measure of the learned $K_i$
>
> We thank the reviewer for the suggestion to strengthen our empirical evidence on the learned representations.
>
> In the revised version, we have added:
> - A table reporting the mean and standard deviation of the learned $K_i$ for each dataset.
> - Figure 14, which visualizes the full distribution of $K_i$​ across graphs.
>
>
> ## Adding Tuning for $K$
> We understand the reviewer’s request for tuning the pooled size $K$ in baseline methods.
>
> First of all, we want to emphasize that competing pooling methods (like MinCut or DiffPool) are constrained to pool every graph to the same size (or ratio) $K$, even if we tune such a value. This is a structural limitation, and it is exactly what BN-Pool addresses by allowing the complexity of the pooled graph to vary with the input.
>
> Moreover, in the experimental comparison, we followed the standard protocol in the literature for competing methods by obtaining the number of clusters $K$ from the data itself (setting $K$ as a fraction of the average number of nodes, $K = 0.5\bar{N}$). This is the standard "rule of thumb" used when $K$ is not treated as a hyperparameter to be grid-searched and **is done in the original papers** proposing the competing methods. In contrast, while we did perform model selection for BN-Pool, the sensitivity analysis suggests that a default configuration would likely have yielded comparable results, mitigating concerns about unfair advantages due to tuning effort.
>
> Nevertheless, we conducted additional experiments to ensure a fair comparison, sweeping over different values of K for soft‑clustering baselines (which are most comparable to our method). The results are reported in Appendix C.
>
> ## Adding Additional Baselines
> Following the reviewer’s recommendation, we expanded our comparison to include several recent and competitive pooling operators:
> - HOSC
> - EigenPool
> - SEP
>
> Due to the computational demands of these additional experiments, a subset of results is still being finalized. The currently available results have been incorporated into the paper and already indicate that our method remains highly competitive across all evaluated benchmarks. Any missing results will be added as soon as possible.

---

### Author Response · Authors · 2026-02-11
**Summary of the Major Revisions**

We thank all reviewers for their thoughtful and constructive feedback. Below we provide a concise recap of the major revisions and improvements implemented in response to the comments:

- **Substantial revision of the methodological presentation.** We reorganized the exposition of graph pooling (sections 2.2 and 2.3) and BN‑Pool (Sections 3.1, 3.2, and 3.3), introduced new figures (Fig. 4, Fig. 5) and revised existing ones (Fig. 7, Fig. 8, Fig. 12, Fig. 14) to improve the presentation. The revised presentation better highlights how the BNP framework and variational inference are framed in the graph-pooling scenario.
- **Introduction of a sparse variant of BN‑Pool.** Although the computational complexity of our original formulation was not a limiting factor in graph‑level tasks and aligned with other Soft-Clustering methods, we developed and tested a sparse version (Sec. 3.4.1 and App. A.4) to further reduce the overhead introduced by the dense computations. This addition increases the practical applicability of BN-Pool to large sparse graphs.
- **More informative training‑time comparisons.** To provide a fairer and more interpretable comparison, we now report (Tab. 8) the number of epochs required by each method. This allows readers to distinguish between per‑epoch computational cost and overall convergence behavior.
- **Expanded set of baselines for graph classification.** We incorporated additional pooling methods (HOSC, EigenPool, and SEP) as suggested by the reviewers to further strengthen the empirical evaluation.
- **Quantitative metrics for the learned $K_i$.** We added Table 4, which shows the mean and standard deviation of the $K_i$ distribution learned by BN-Pool across datasets.
- **New experiments with tuned K for soft‑clustering baselines.** To ensure a fair comparison, we ran additional experiments in which soft‑clustering baselines were evaluated across multiple values of K. Results are in Appendix C.
- **Addition of a multipartite dataset to illustrate a challenging scenario.** To demonstrate the limitations of our method and provide a more balanced evaluation, we included results on the Multipartite dataset where the structure is not aligned with the generative assumptions of BN‑Pool.

We believe these changes significantly improve the clarity, completeness, and empirical strength of the paper, and we thank the reviewers once again for the valuable suggestions.
All modifications introduced in response to the reviews are highlighted in magenta in the revised PDF to assist the reviewers in locating the changes.

---

> ### Author Response · Authors · 2026-02-18
>
> We have completed all the additional experiments and updated the PDF accordingly.
> Since today marks the final day of the rebuttal period, please let us know if there are any remaining points that would benefit from clarification.

---

### Decision · Action_Editor_9H9E · 2026-03-16

**Recommendation:** Accept as is

**Audience:**

Yes

**Audience Explanation:**

Yes. This manuscript concerns several topics that are individually of interest to substantial subsets of the TMLR audience: Bayesian nonparametrics, graph neural networks, and clustering.

**Claims And Evidence:**

Yes

**Claims Explanation:**

During the initial review period, some minor concerns were raised by the reviewers, which were adequately resolved via revision / discussion with the authors.

The reviewers now universally agree that the claims made in the (updated) manuscript are adequately supported.